# Social sentiment segregation: Evidence from Twitter and Google Trends in Chile during the COVID-19 dynamic quarantine strategy

**Fernando Díaz**[1,2], **Pablo A. Henríquez**[1,2] *

**1** Departamento de Administración, Facultad de Economía y Empresa, Universidad Diego Portales, Santiago, Chile, **2** Centro de Investigación Empírica en Negocios, Facultad de Economía y Empresa, Universidad Diego Portales, Santiago, Chile

☯ These authors contributed equally to this work.
* pablo.henriquez@udp.cl

**Data Availability Statement:** All relevant data are within the paper and its Supporting information files.

## Abstract

The Chilean health authorities have implemented a sanitary strategy known as *dynamic quarantine* or *strategic quarantine* to cope with the COVID-19 pandemic. Under this system, lockdowns were established, lifted, or prolonged according to the weekly health authorities' assessment of municipalities' epidemiological situation. The public announcements about the confinement situation of municipalities country-wide are made typically on Tuesdays or Wednesdays before noon, have received extensive media coverage, and generated sharp stock market fluctuations. Municipalities are the smallest administrative division in Chile, with each city broken down typically into several municipalities. We analyze social media behavior in response to the confinement situation of the population at the municipal level. The dynamic quarantine scheme offers a unique opportunity for our analysis, given that municipalities display a high degree of heterogeneity, both in size and in the socioeconomic status of their population. We exploit the variability over time in municipalities' confinement situations, resulting from the dynamic quarantine strategy, and the cross-sectional variability in their socioeconomic characteristics to evaluate the impact of these characteristics on social sentiment. Using event study and panel data methods, we find that proxies for social sentiment based on Twitter queries are negatively related (more pessimistic) to increases in the number of confined people, but with a statistically significant effect concentrated on people from the wealthiest cohorts of the population. For indicators of social sentiment based on Google Trends, we found that search intensity during the periods surrounding government announcements is positively related to increases in the total number of confined people. Still, this effect does not seem to be dependent on the segments of the population affected by the quarantine. Furthermore, we show that the observed heterogeneity in sentiment mirrors heterogeneity in stock market reactions to government announcements. We provide evidence that the observed stock market behavior around quarantine announcements can be explained by the number of people from the wealthiest segments of the population entering or exiting lockdown.

**Funding:** The author(s) received no specific funding for this work.

**Competing interests:** The authors have declared that no competing interests exist.

## 1 Introduction

The COVID-19 pandemic that began in Wuhan, China, in December 2019 [1] rapidly spread to the rest of the globe during 2020, reaching unprecedented proportions. As of 17 March 2021, 119,960,700 cases have been confirmed, 2,656,822 deaths and 326,858,656 vaccine doses have been administered [2]. COVID-19 has become an epidemiological and economic global crisis. [3].

The Chilean government declared a state of health emergency on Feb 8th, nearly a month before March 3rd when the first case of coronavirus was detected in Chile [4]. In spite of this early government response to the pandemic, the economic and social effects of the global crisis have been severe. On Monday, March 16th, the Selective Stock Price Index of the Santiago Stock Exchange (*IPSA*), which comprises the 40 most heavily traded stocks, plunged by 14.11% reaching 3,232 points after the government announced the closing of the borders in order to curb the expansion of the coronavirus. At the same time, the General Stock Price Index (IGPA) fell from 18,896 points to 16,454 points, an almost 13% drop in value (https://www.bolsadesantiago.com). A decrease in the employment rate of around 20% towards the end of 2020 resulted in an increase in the unemployment rate and a noticeable drop in the country's workforce (https://www.ine.cl). As of December 2020, one third of the total workforce was unemployed. Around 70% of those workers had their work contracts on hold. Formal and informal employment figures also declined, the fall mainly affecting women. There was a vast increase in personal debts affecting households as well as small businesses, which resulted in many of them closing down. As of late March 2021, the number of people diagnosed with the disease reached almost one million, with a total of approximately 23,000 deaths.

In this context of high uncertainty and fear about the sanitary crisis's consequences, social media has played a key role in the contagion and transmission of information about the pandemic. The amount of information disseminated through social networks has reached levels rarely seen before. Kumar et al [5] report how Twitter has emerged as a critical tool for communicating the effects of this crisis and report that during its early stages, there was a COVID-19-related tweet every 45 ms. According to these authors, *"A social media pandemic has preceded the disease pandemic, stirring a diversified spectrum of emotions"*. Mavragani and Gkillas [6] analyze the role of Google query data in the predictability of COVID-19 and show evidence for a significant correlation between Google Trends and COVID-19 data in the United States.

The Chilean health authorities implemented a sanitary strategy known as a *dynamic quarantine* or *strategic quarantine* to cope with the COVID-19 pandemic. Under this system, *lockdowns* are implemented every week in some municipalities and lifted from others, according to the health authorities' assessment of their epidemiological situation [7]. The decisions are made considering different factors, including the number of new cases in a given municipality, the size of its elderly population, and the access of its inhabitants to health care. The corresponding public announcements, typically made on Tuesdays or Wednesdays before noon, receive extensive media coverage and produce large fluctuations in the stock market.

The primary aim of this paper is the analysis of social media behavior in response to the measures taken by the Chilean government regarding lockdowns. The *dynamic quarantine* scheme constitutes a unique opportunity to assess the impact of confinement on social sentiment. Given that municipalities differ in size as well as socioeconomic status of their population, as they alternate between being or not being in lockdown, it is possible to assess the impact of these characteristics on social sentiment. By classifying the population according to the socioeconomic status *(SES)* of the municipalities in which they live, we provide evidence of heterogeneity in the responses of social sentiment to the lockdown announcements.

Furthermore, we document that the observed heterogeneity in sentiment responses mirrors heterogeneity in stock market reactions to government announcements. We find statistically significant stock market reactions to lockdown announcements, whose magnitude and signs are related to the number of people affected by such announcements and with their economic significance largely being concentrated in the country's wealthiest population. This result is important for our analysis, because it validates the *SES* based segmentation we entertain in our sentiment analysis.

For social media analysis, we resort to Twitter queries to compute a sentiment index [8–10] as well as Google Trends to compute a search index intensity of specific words related to the pandemic [11, 12]. We find that our Twitter-based sentiment proxy is negatively related (more pessimistic) to increases in the number of people under lockdown, but with a statistically significant effect only for changes in the numbers under lockdown from the wealthiest cohorts of the population. This suggests the existence of socioeconomic segregation among users of this platform. Concerning Google searches, we find that search intensity during periods surrounding government announcements is positively related to changes in the total number of people under lockdown, with little or no evidence of socioeconomic segregation.

We contribute to the existing literature in several ways. Firstly, we provide evidence that the observed heterogeneity in Twitter-based sentiment responses is closely related to the *SES* of the population under lockdown, with a discernible effect concentrated in the wealthiest cohorts. This is an important issue both for academic research and policymakers. If observable social sentiment variables are used to assess the impact of policy measures on the overall well-being of the population, it is possible that the observed effects reflect the feelings of only part of the population towards those measures. There are only a few papers that have investigated the causal effect between lockdowns and overall population well-being [13–15]. However, to the best of our knowledge, we are the first to take advantage of the heterogeneity on social sentiment resulting from *dynamic quarantines*. Furthermore, our empirical setup constitutes a novel approach that allows extracting the socioeconomic status of users of social network platforms [16]. Secondly, and from a methodological point of view, the high degree of intracity socioeconomic segregation that the country exhibits, together with the characteristics of the *dynamic quarantine* scheme, allows us to investigate the effects of the quarantine announcements on market sentiment at the smallest administrative level in Chile, which allows for the construction of better counterfactuals for our analyses [4]. Finally, to validate our socioeconomic sorting criterion, we relate stock market reactions to the size and socioeconomic characteristics of the population under lockdown. We find that government announcements produce significant stock market reactions, but only when the wealthiest municipalities are involved in such announcements.

The rest of the paper is structured as follows. The next section contains a brief review of related literature. Section 3 describes our data and research methods. Section 4 presents our main results. The paper concludes in section 5.

## 2 Literature review

The extant literature provides extensive evidence of the relationship among stock price and economic variables behavior, market-wide sentiment, and people's interactions in social media. Table 1 presents a tabular format of some of the recent papers on sentiment analysis. In a seminal paper, Baker and Wurgler [17] show that investor sentiment has discernible and regular effects on individual firms and the stock market as a whole. Bijil et al [18] find evidence that Google search volumes can predict stock returns, with high search volumes leading to negative returns. Azar an Lo [19] argue that the content of tweets related to the Federal Open

**Table 1. Summary of methods applied in related literature.**

| Reference | Findings and conclusions |
|---|---|
| [18] | The relationship between Google search volumes and stock return changes over time. |
| [19] | Argues that the content of tweets related to the Federal Open Market Committee meetings in the US can be used to predict future returns. |
| [20] | Uses Social media Twitter messages to construct sentiment measures and shows that stock market prices react to both firm-specific and market-wide sentiment. |
| [22] | The results show that Twitter sentiment provides new information on analysts' recommendations, analysts' price targets and quarterly earnings. |
| [23] | Introduced an economically motivated model for using Google search frequency data to forecast volatility |
| [24] | A state-of-the-art sentiment classification technique in order to investigate the question of whether sentiment and attention measures contain additional predictive power for realized volatility when controlling for a wide range of economic and financial predictors. |
| [25] | The Google searches can both explain and predict trading volume. |
| [30] | This paper deals with the sentiment analysis of Indians after the lockdown announcements were made. It can be seen that Indians have received the fight against COVID-19 positively and the majority are in agreement with the government. |
| [31] | Studies about sentiment analysis in the presence of infectious diseases, outbreaks, epidemics and pandemics over a 10-year period were systematically reviewed. |
| [32] | The present study applied sentiment analysis on Twitter data related to worldwide COVID-19 outbreaks. They can say that people's reactions vary day-to-day from the postings on social media, specifically Twitter. |
| [33] | They propose a model to deploy a Gaussian fuzzy-rule based technique to evaluate the sentiments expressed in tweets. |
| Present paper | Provides evidence that social sentiment based on Twitter queries tends to be more pessimistic the higher the proportion of the wealthiest population under lockdown is. This socioeconomic segregation in social media is also observed in the financial market, in which stock market abnormal returns can be explained by the number of people from the wealthiest segment of the population that enters or exits confinement. |

Market Committee meetings in the US can be used to predict future returns. Broadstock and Zhang [20] use Twitter messages to construct sentiment measures and show that such measures carry pricing power against the stock market. Preis et al [21] analyze changes in finance-related Google Trend query volumes and find evidence that these changes might be able to anticipate future trend patterns. Gu [22] find that Twitter sentiment predicts stock returns without subsequent reversals and argue that this finding provides evidence consistent with the view that Twitter messages contain information not reflected in stock prices.

Concerning market volatility, Hamid and Heiden [23] forecast volatility in stock markets using Google search frequency as a measure of investor attention, finding that prediction accuracy increases together with investor attention during highly volatile periods. Audrino and Ballinari [24] find that sentiment and attention variables have significant predictive power for stock market volatility. Kim et al [25] find that increased Google searches predict increased volatility and trading volume.

The COVID-19 pandemic has been a fertile ground for analyzing the relationship between economic and financial variables and social sentiment. Lyócsa et al [11] show that fear of coronavirus, proxied by excess Google search volumes, predicts price variation during the pandemic stock market crash. van der Wielen and Barrios [26] present evidence from country-specific internet searches of a substantial change for the worse in people's economic sentiment in the months following the coronavirus outbreak. Lyócsa and Molnár [12] use a nonlinear autoregressive model to analyze stock price autocorrelation in the *SP500* index. The transition variables are the abnormal Google searches related to COVID-19, and the market realized

volatility. They find that the autocorrelation of market returns increased in magnitude and remained negative in periods of extreme market volatility and when attention to COVID-19 increased.

More closely related to our work, Greyling et al [13] find that the lockdowns in South Africa have had a significant and negative impact on people's happiness. Greyling et al [14] analyze the causal effect of mandatory lockdowns on happiness in South Africa, New Zealand, and Australia and find that lockdowns negatively affect happiness. Brodeur et al [15] analyze whether the lockdowns implemented in Europe and America led to changes in population well-being. The authors suggest that people's mental health may have been severely affected by the lockdown.

Finally, and related to the literature on the socioeconomic status inference of social media hidden user characteristics, which is one of the most active information retrieval fields, the following references are worth mentioning. Ai et al [27] evaluate the inference accuracy gained on latent attribute inference models by augmenting the user characteristics with features derived from the Twitter profiles and postings of friends. Filho et al [28] propose a method to automatically generate a user social class, taking advantage of Foursquare user interactions and Twitter messages. Volokova et al [29] propose an approach to predict latent personal attributes, including user demographics, online personality, emotions, and sentiments from texts published on Twitter.

## 3 Methods

### 3.1 Data

All the databases were constructed for the sample period between January and August 2020. All the data were collected according to the Terms of Use and Service of the source websites and are made available in (S1 File).

### 3.2 Lockdowns and population *SES*

There are two distinct stages in the pandemic strategy of the Chilean government. In the first stage, corresponding to the period between March 24 and July 20, the government imposed complete lockdowns in different municipalities according to their pandemic situation. In the second stage, with the so-called *step by step plan* announced on July 19th, the government changed its strategy to ease the complete lockdowns imposed up to that point. The plan is a gradual pandemic strategy according to each area's health situation, but with five stages or incremental steps, ranging from Quarantine, a step equivalent in stringency to the previous stage lockdowns, to Advanced Opening.

We restrict our attention to the first stage, in which the health authorities made a total of 25 weekly announcements about what municipalities nationwide could be under lockdown during the following week, available at https://www.minsal.cl/. We consider the confinement situations of all municipalities, with a population larger than 13,000 people. This sample covers 120 municipalities, approximately 14,500,000 people, which represents 83% of the country's total population. We obtain the municipalities' populations from the 2017 census reported by the *National Statistics Institute (Instituto Nacional de Estadísticas, INE)*, available at https://www.ine.cl/. As a proxy for the *SES* of the population, we resort to the *Multidimensional Poverty Index (MPI)* reported by the *Ministry of Social Development (Ministerio de Desarrollo Social)* in the CASEN 2017 survey, available at http://observatorio.ministeriodesarrollosocial.gob.cl. The *MPI* is an international measure of acute multidimensional poverty. It complements traditional monetary poverty measures by capturing the acute deprivations in health, education, and living standards that a person faces simultaneously. For each week in our

**Table 2. MPI descriptive statistics by SES.**

|  | % Total Population | MPI (Average) | Variability (Average) |
|---|---|---|---|
| Wealthiest 5 Mun. | 4.68% | 0.53% | 0.73% |
| Wealthiest 10 Mun. | 9.57% | 1.53% | 1.23% |
| Wealthiest 15 Mun. | 12.76% | 2.26% | 1.92% |
| Wealthiest 20 Mun | 18.20% | 2.77% | 2.08% |
| Wealthiest 24 Mun. | 22.37% | 3.07% | 2.17% |
| Total Sample | 83.00% | 8.49% | 4.38% |

sample period, and according to the government's announcements about the municipalities that will be under lockdown, we obtain the approximate total number of people that will be under lockdown until the next announcement, considering the population of the municipalities entering and leaving quarantine. To get the number of people from the wealthiest population that are in lockdown each week, we proceed as follows. We sort the 120 municipalities in our sample according to their *MPI*. Starting from the wealthiest municipality, i.e. the one with the lowest *MPI*, we add municipalities until we accumulate approximately 12% of the total population of the country, a percentage that corresponds to the proportion of people belonging to the *ABC*1 socioeconomic segment for 2018, according to the *Association of Market Researchers and Public Opinion of Chile (Asociación de Investigadores de Mercado y Opinión Pública de Chile, AIM,* https://www.aimchile.cl/). We end up with 15 municipalities that we consider include the country's wealthiest population, comprising 2,136,062 inhabitants.

Since lockdowns affect all the inhabitants of a given municipality, it should be noted that to obtain the approximate number of people from a given SES that is confined upon government announcements, we assume that the whole population of that municipality belongs to the same segment, disregarding the SES heterogeneity that their inhabitants naturally have. However, a quick look at Table 2 reveals that wealthy municipalities exhibit a much lower variability in the MPI of their inhabitants than non-wealthy ones, where variability is defined as the range of the poverty index for a given municipality. In this sense, since wealthy municipalities are far less heterogeneous, identifying the wealthiest population through the municipality they reside in does not seem particularly troublesome.

In S1 Table, we present a list of the municipalities included and their population, sorted according to the *MPI*.

### 3.3 Stock market reactions

We resort to standard tools in the *event study methodology* to assess the impact of lockdown announcements on the stock market [34–36]. Daily data on stock market indexes used in the analysis- *IPGA, IPSA, S&P500, Dow Jones Industrial*—were downloaded from *investing.com.* Fig 1 shows the evolution of the *IGPA* index and its volatility throughout our sample period. Like most stock markets globally, the Chilean index has experienced sharp swings during the pandemic and exhibits historically high levels of volatility.

We consider the 25 government lockdown announcements made from March 24 to July 20, the period previous to the so-called *Step by Step Program* implemented by the government in late July. We compute abnormal returns for the *IGPA* and *IPSA* indexes by deducting expected returns, predicted by the *Market Model*, from actual returns of the corresponding index. We use both the S&P 500 index and the Dow Jones Industrial index as benchmark portfolios. The estimation window for the market model ranges from January 1st, 2020, to

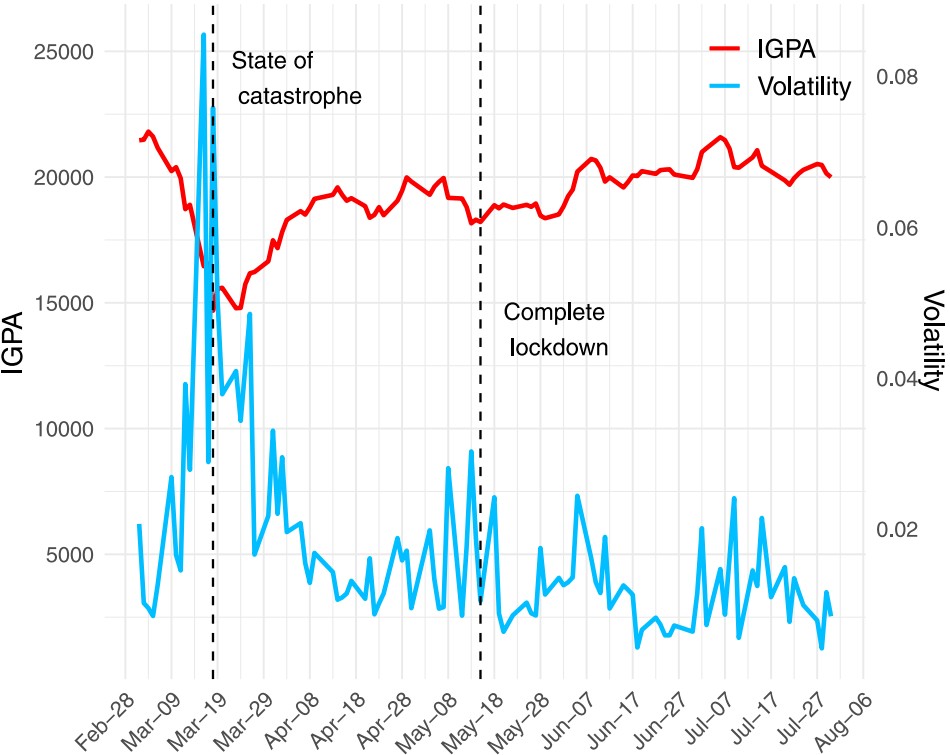

**Fig 1. IGPA index and its volatility.** Chilean market and volatility from March 2020 to end of July 2020. Following Chou et al [37], the volatility estimator of the *IGPA* index is computed as $\sigma = ln(High/Low)/\sqrt{4ln(2)}$, where *High* and *Low* corresponds to the highest and lowest observed level of the index on any given day.

February 15th, 2020. The end date of the estimation window was chosen so that the pandemic's early consequences on security prices did not influence the expected return estimates.

The market model estimation is performed by a simple *OLS* regression of the form,

$$R_{i,t} = \alpha_i + \beta_i R_{m,t} + \epsilon_{i,t} \tag{1}$$

where *i* denotes the specific domestic index being considered, $R_m$ denotes the return of the benchmark portfolio, *t* is time, and $\epsilon_{i,t}$ is an error term with expectation zero, finite variance and uncorrelated to the return of the benchmark portfolio. We tried different expected returns models. As expected, and given the short time horizon of our analysis, our results are qualitatively unchanged [38].

Abnormal returns are then computed as,

$$AR_{i,t} = R_{i,t} - (\hat{\alpha}_i + \hat{\beta}_i R_{m,t}) \tag{2}$$

where $\hat{\alpha}_i$ and $\hat{\beta}_i$ are the OLS estimates of Eq (1).

Since government announcements were made almost weekly, we considered short windows around announcement days for our analysis, so that the estimated effect of one announcement is not influenced by the effect of the previous announcement nor does it influence the effect of the next announcement on stock prices. As pointed out by Kothari and Warner [38], short-horizon methods are quite reliable. Given that the government announcements are typically made before noon, market reactions on the same day of the announcements are particularly relevant. We therefore consider two *event windows*. The first one is the (−1, 0) window, which

includes the day before the announcement, −1, and the same day of the announcement, 0. The second one is the (−1, +1) window commonly used in short horizon event studies, which further includes the day after the announcement, +1. Individual abnormal returns at day $t$ are added up inside the corresponding event window to compute cumulative abnormal returns for that window,

$$CAR_i(t_1, t_2) = \sum_{t=t_1}^{t_2} AR_{i,t} \tag{3}$$

To test the null hypothesis that the mean abnormal performance equals zero for a specific announcement, the standard approach is to compute the following test statistic,

$$z = \frac{CAR(t_1, t_2)}{\sqrt{\sigma^2(t_1, t_2)}} \tag{4}$$

where $\sigma^2(t_1, t_2)) = L\sigma^2(AR_t)$, $\sigma^2(AR_t)$ is the variance of the one-day abnormal return and $L = t_2 - t_1$ is the length of the event window. This test statistic is typically assumed unit normal in the absence of abnormal performance [39].

We then relate the observed stock market reactions to the number of people being quarantined and their socioeconomic characteristics. We use the statistic in Eq 3 as the dependent variables in a regression set up in which the independent variables are different cohorts of the population under lockdown based on their socioeconomic characteristics:

$$CAR_t = \beta_1 + \beta_2 \Delta Population_{j,t} + \mu_t \tag{5}$$

where $\Delta Population_{j,t}$ is the change in the number of people from cohort $j$ confined at the announcement made at time $t$.

## 3.4 Market sentiment

Our first sentiment proxy is based on Twitter queries. We collect data for non-protected users using the API provided by Twitter. As part of the data gathering process, all potentially relevant tweets, filtered using the hashtags #COVID2019chile and #CoronaVirusEnChile, were searched and extracted from Twitter using the **twitteR** package. The final dataset contains a collection of 1,214,564 tweets related to COVID-19 in Chile during our sample period. The raw data, having polarity, is highly susceptible to inconsistency and redundancy. Pre-processing of the tweets includes the removal of all URLs, punctuation, numbers, and other like symbols. After the pre-processing stage, each tweet is then labeled as positive or negative, based on a list of approximately 700 English positive and negative *opinion related words* or sentiment related words that we translate into Spanish from Hu and Liu [10]. We then assess the sentiment polarity of each tweet using a *Sentiment Score*, which determines the direction of the sentiment as well as its strength [40, 41].

$$Sentiment\ Score = \frac{positive - negative}{positive + negative + 2}, \tag{6}$$

where *positive* (*negative*) represents the positive (negative) words count. Accordingly, the *Sentiment Score* falls into the range [−1, s1]. Since the *Sentiment Score* ranges from −1 to 1, we first compute a *Normalized Sentiment Score, (NSS)*, by scaling data using a *Min-Max*

*normalization*:

$$NSS_t = \frac{Sentiment\ Score_t - Sentiment\ Score_{min}}{Sentiment\ Score_{max} - Sentiment\ Score_{min}} \quad (7)$$

The *NSS* ranges from 0 to 1. Our goal is to compute an *Abnormal Sentiment Activity* index susceptible to be tested for statistical significance around quarantine announcement days. To this end, we follow Da et al [42] and define an *Abnormal Sentiment Activity* in day *t* as:

$$ASA_t = ln\left(\frac{NSS_t}{med\{NSS_{t-1}, \ldots, NSS_{t-5}\}}\right) \quad (8)$$

where *ln* denotes natural logarithm. $ASA_t$ can be considered the change between the current normalized sentiment score, $NSS_t$, and the median (*med*) of such measure over the previous five trading days. The use of five days instead of the previous day to compute the abnormal sentiment activity is used address the potential noisiness of daily market sentiment measures, as proposed by [11, 12]. Furthermore, the choice of five days allows us to compute a normal sentiment level that should not be affected by previous government announcements, since such announcements are typically made six or seven days apart. The evolution of the *Sentiment Score*, the *Normalized Sentiment Score* and the *Abnormal Sentiment Activity* index for our sample period is shown in Fig 2. For a better visualization, in Fig 2A, we show the most negative point, resulting from the announcement on May 13th of the complete lockdown of the whole Metropolitan area of Santiago, in a subplot.

Our second sentiment measure is based on Google Trends. As described in Nagoa et al [43], Google Trends (GT) is a service that outputs the time series data of search intensity to show the extent to which a particular keyword is searched for in a specified period and location. The intensity is measured in a scale that ranges from 0 to 100, where the value of 100 indicates the peak of popularity (100% of popularity in given period and location) and 0 (complete disinterest). GT may qualify analyzed phrases as either search term or topic. Search terms are literally typed words, while topics may be proposed by GT when the tool recognizes phrases related to popular queries.

We retrieve data on the search volume intensity of the following 19 terms that are specifically related to the virus outbreak and subsequent policy interventions: corona, OMS (WHO), virus, COVID-19, SARS, MERS, epidemia (epidemic), pandemia (pandemic), síntoma (symptom), infectado (infected), propagación (spread), brote (outbreak), distanciamento social (social distancing), restricción (restriction), cuarentena (quarantine), suspender (suspend), viajar (travel), encierro (lockdown) and mascarilla (face mask). We specify the region as *CL* (Chile).

We aggregate search intensity across terms mentioned above by taking the average across all individual indices for each day *t*. The result is the *Average Search Volume Intensity* index, $ASVI_t$. The higher the value of the $ASVI_t$ on a given day *t*, the higher the population's attention to the outbreak of Coronavirus on that day. To study how changing patterns in search activity are related to market uncertainty, we follow the work of Da et al [42] and calculate the *Abnormal Search Volume Activity*, $ASVA_t$:

$$ASVA_t = ln\left(\frac{ASVI_t}{med\{ASVI_{t-1}, \ldots, ASVI_{t-5}\}}\right) \quad (9)$$

where *ln* denotes natural logarithm. $ASVA_t$ can be interpreted as the change between the current search volume intensity $ASVI_t$ and the median (med) over the previous five trading days. The use of five days instead of the previous day to compute the search in volume activity is motivated by the potential noisiness of search volume intensities, as proposed by [11, 12]. The

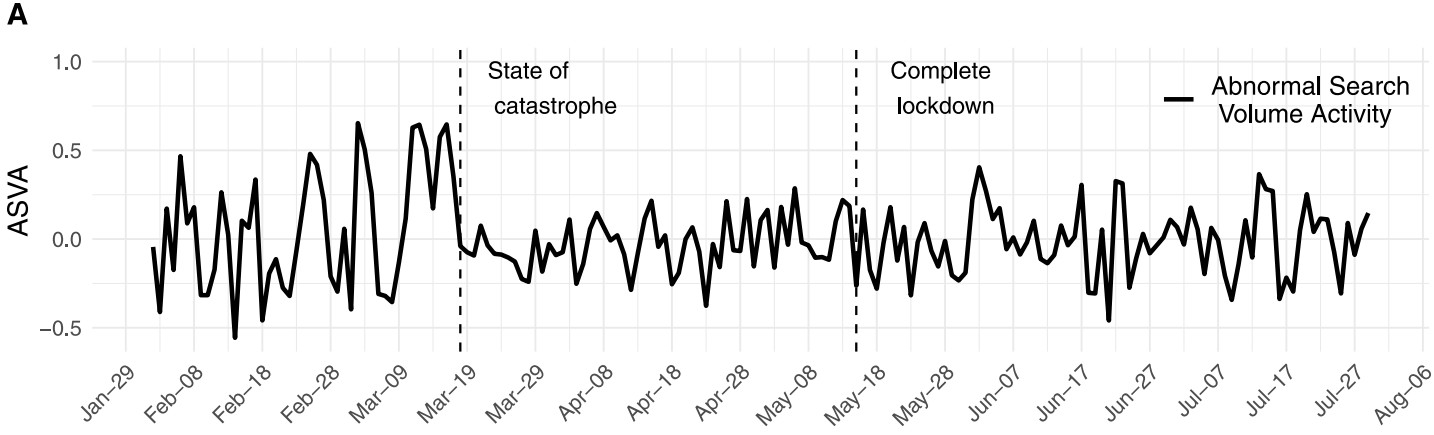

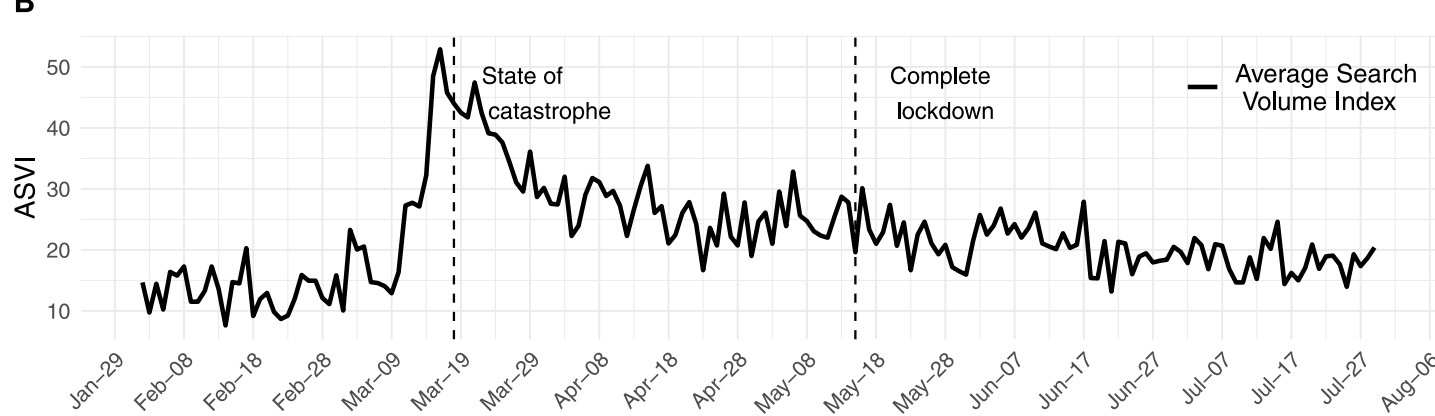

**Fig 2. Time series of sentiment of Twitter related to COVID-19.** A) Abnormal Sentiment Activity, B) Sentiment score is normalized between 0 and 1. Regarding the normalization of the *Sentiment Score*, values close to 0 indicate strong negative sentiment while values close to 1 indicate positive sentiment. C) Sentiment Score into the range [−1, 1].

evolution of the $ASVI_t$ and the $ASVA_t$ series for the period between January and August, 2020, is shown in Fig 3.

Having obtained our two market abnormal sentiment proxies for each day $t$ in our sample, $ASA_t$ and $ASVA_t$, we consider two alternative empirical approaches to analyze the effect of lockdown announcements on market sentiment.

### 3.5 Event study methods for sentiment analysis

As a first approach, and analogously to the standard practice in the event study methodology for stock returns, we add up the abnormal sentiments indexes, $ASA_t$ and $ASVA_t$, defined in Eqs 8 and 9, respectively, inside the event windows for each of the 25 lockdown announcements in our sample to compute cumulative abnormal sentiment-related variables. One advantage of this approach is that it allows us to test directly whether such announcements produce a statistically significant effect on social sentiment. For our Twitter based sentiment index, we define a *Cumulative Abnormal Sentiment Activity* statistic, *CASA*:

$$CASA_i(t_1, t_2) = \sum_{t=t_1}^{t_2} ASA_{i,t}, \tag{10}$$

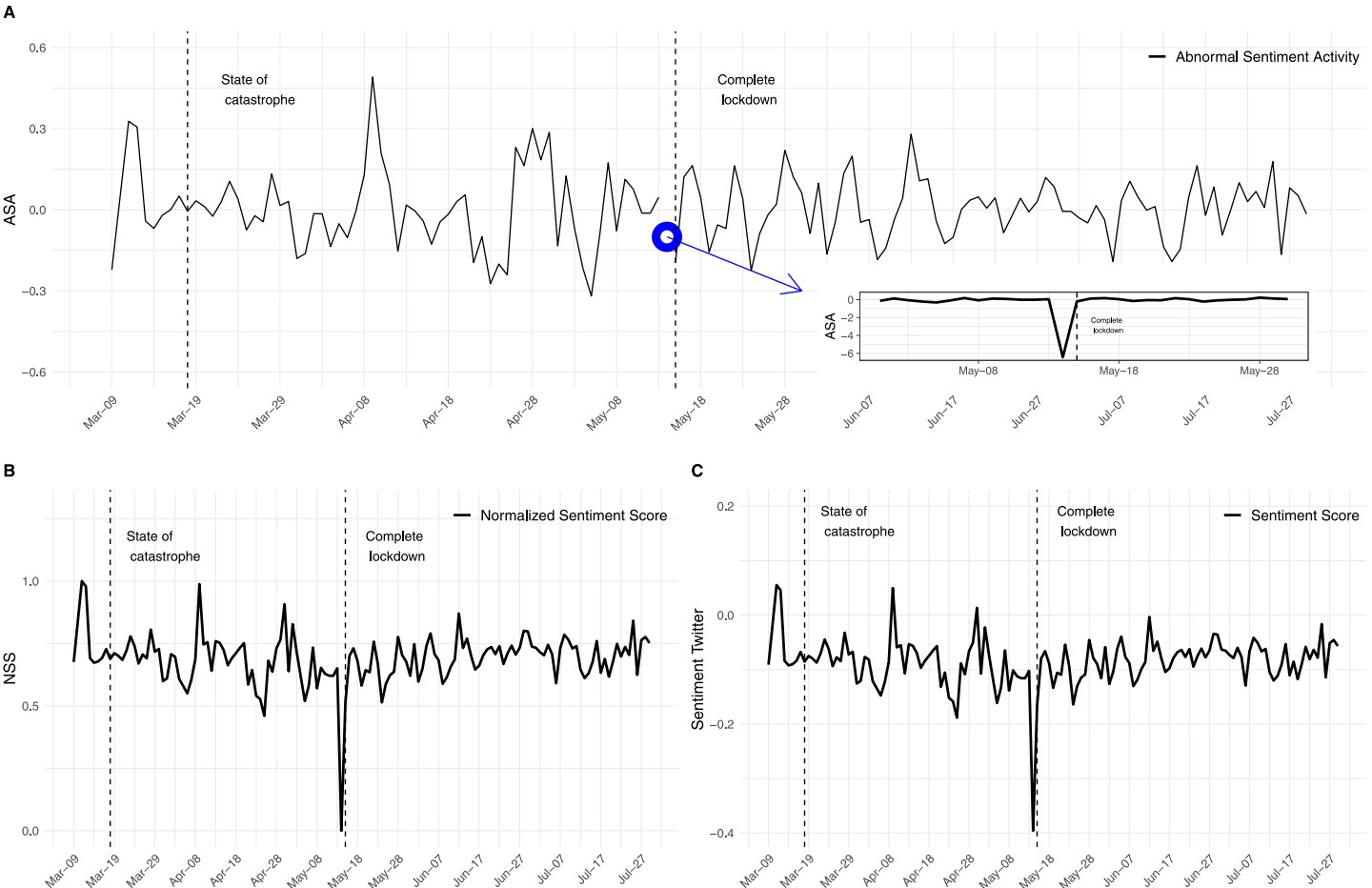

**Fig 3. The evolution of the $ASVA_t$ and the $ASVI_t$ series.** A) Abnormal Search Volume Activity and B) Average Search Volume Intensity. Data obtained using **gtrendsR** R package. Values below 1, denoted as "< 1", are replaced by 0.

and we test its statistical significance using the following statistic,

$$z = \frac{CASA_i(t_1, t_2)}{\sqrt{\sigma^2(t_1, t_2)}} \tag{11}$$

where $\sigma^2(t_1, t_2)) = L\sigma^2(ASA_t)$, $\sigma^2(ASA_t)$ is the variance of the one-day abnormal sentiment activity, and $L = t_2 - t_1$ is the length of the event window. To assess the impact of government lockdown announcements on the search volume intensity, we compute a *Cumulative Abnormal Search Volume Activity*, *CASVA*:

$$CASVA_i(t_1, t_2) = \sum_{t=t_1}^{t_2} ASVA_{i,t} \tag{12}$$

To test for significance, we use the following statistic,

$$z = \frac{CASVA_i(t_1, t_2)}{\sqrt{\sigma^2(t_1, t_2)}} \tag{13}$$

where $\sigma^2(t_1, t_2)) = L\sigma^2(ASVA_t)$, $\sigma^2(ASVA_t)$ is the variance of the one-day abnormal search volume activity and $L = t_2 - t_1$ is the length of the event window.

Analogous to what we do with cumulative abnormal returns for the stock market, we use the statistics in Eqs 10 and 12 as dependent variables in a regression setup in which the independent variables are different cohorts of the population based on the socioeconomic characteristics of the municipalities under lockdown at a specific announcement made at time $t$:

$$CASA_t = \beta_1 + \beta_2 \Delta Population_{j,t} + \gamma x_t + \mu_t \qquad (14)$$

$$CASVA_t = \beta_1 + \beta_2 \Delta Population_{j,t} + \gamma x_t + \mu_t \qquad (15)$$

where $CASA_t$ ($CASVA_t$) is the *Cumulative Abnormal Sentiment Activity* (*Cumulative Abnormal Search Volume Activity*) statistic for the quarantine announcement made at time $t$, $\Delta Population_{j,t}$ is the change in the number of people from cohort "$j$" confined at the announcement made at time $t$, and $x_t$ is a vector of controls related to stock market performance and country-wide pandemic conditions.

## 3.6 Panel methods for sentiment analysis: A DiD "like" estimator

As a second approach, to take advantage of the panel structure of our data in which municipalities with different socioeconomic characteristics enter and exit lockdowns periodically, a natural way to proceed is to consider a *Difference in Difference (DiD)* methodology to analyze the effect of lockdowns on social sentiment [44–48]. Following the approach in recent literature [42], we consider the $ASA_t$ and $ASVA_t$ indexes as our outcomes of interest.

Given the nature of the dynamic quarantine scheme, we have different treatment timings for different municipalities, a setup that is sometimes referred to as a *staggered DiD* model [49, 50]. It should be noted that the standard *DiD* estimator, defined as the difference in average outcome in the treatment group, before and after treatment, minus the difference in average outcome in the control group, before and after treatment, is not feasible in our case. We observe our outcome of interest for the population as a whole, without distinction of socioeconomic status. In other words, for each day $t$ in our sample period, we observe a single value of either $ASA_t$ or $ASVA_t$.

Despite the fact that we observe a unique value of the sentiment indexes for both the treatment and control groups, it is still possible to compute an estimator in the spirit of a *DiD* estimator. To see this, let $y_t$ be the observable outcome, common to both groups. Consider the $k^{th}$ government announcement made at day $t^k$ and the event window $(t_1^k, t_2^k)$, centered around that announcement day. Let $Q_t$ be an indicator variable that takes the value of 1 if $t_1^k \leq t \leq t_2^k$, and zero otherwise; i.e., $Q_t$ indexes all calendar days $t$ that fall within an event window around a government announcement. Let $I_{ik}$ be an indicator variable with the value of 1 if municipality $i$ is announced to go into lockdown at government announcement $k$, and zero otherwise. Consider the $D_{it}^{(j)}$ indicator variable defined as follows;

$$D_{it}^{(j)} = \begin{cases} 1 & \text{if } I_k = 1 \text{ and } Q_{t+j} = 1 \\ 0, & \text{otherwise} \end{cases} \qquad (16)$$

Accordingly, variable $D_{it}^{(j)}$ takes a value of 1 for calendar days that are $j$ days apart from $t^k$ and belong to the event window centered around $t_k$, for all municipalities $i$ that were locked down at the corresponding announcement. Let $W_i$ be an indicator variable that takes the value

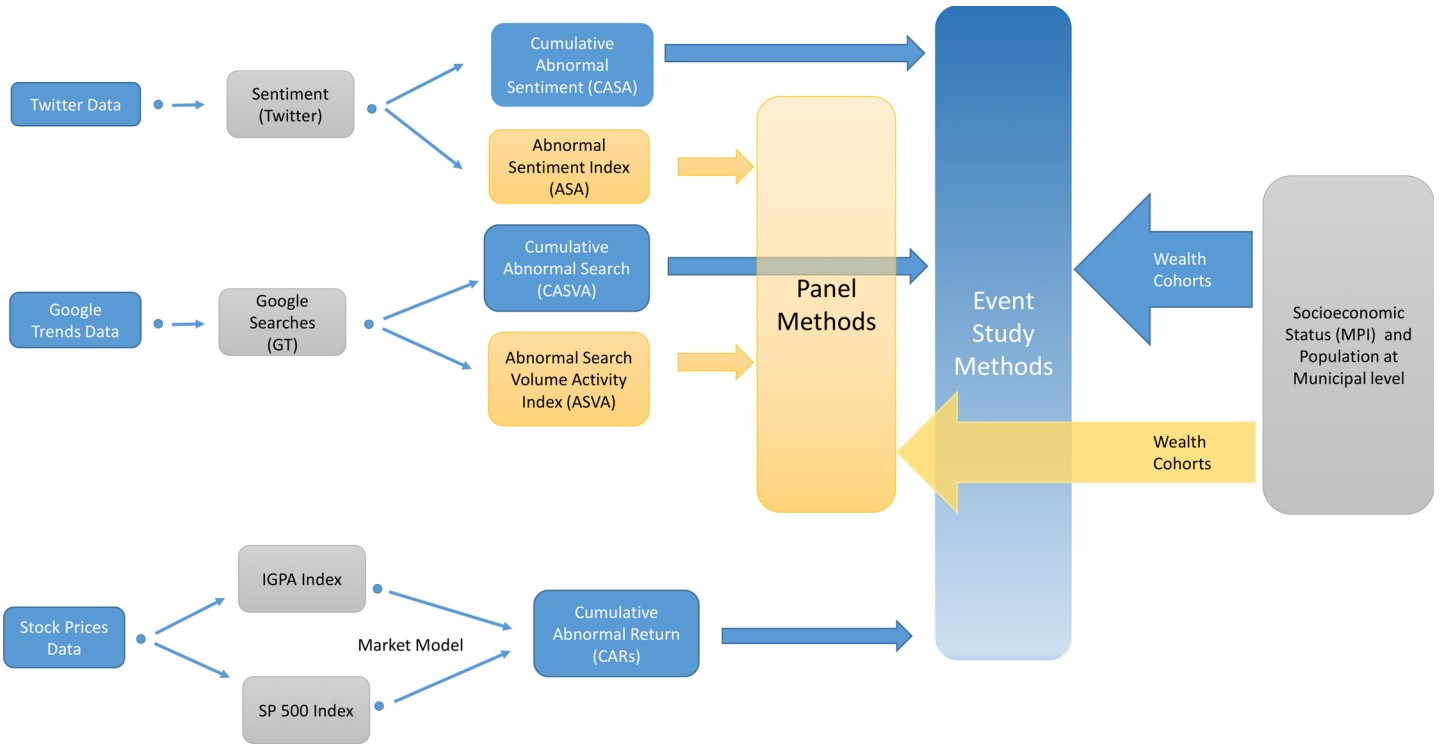

**Fig 4. Workflow diagram of the proposed empirical methodology.**

of 1 if municipality $i$ is wealthy and zero otherwise and consider the following specification:

$$y_t = \alpha_1 + \sum_{l=-j}^{l=+j} \delta_l D_{it}^{(j)} + \sum_{l=-j}^{l=+j} \beta_l D_{it}^{(j)} \cdot W_i + \gamma W_i + \mu_{it} \tag{17}$$

The parameters $\beta_l$ are similar to the standard $DiD$ estimators, but they consider only time variation in the outcome of interest. They correspond to the average difference in $y_t$ between the effects of being or not being under lockdown, for wealthy versus non-wealthy municipalities, $l$ days before (if $l < 0$) or after (if $l > 0$) a quarantine announcement.

In Fig 4 we show diagrammatically the workflow diagram of our proposed methodology, from the data gathering process and variable creations to the formal empirical models.

## 4 Results

### 4.1 Stock market reactions and sentiment responses to lockdown announcements

In columns (1) and (2) in Table 3, we report the stock market reactions to each of the government announcements relating to the dynamic quarantine scheme. In column (1), we report the *IGPA* index cumulative abnormal returns (*CAR*) for the $(-1, 0)$ window. In column (2) we report the observed *CARs* for the $(-1, +1)$ window. Since our results are qualitatively unchanged using any of the domestic indexes and either choice of the benchmark portfolio, we only report our results for the *IGPA* index using the *S&P500* as the benchmark portfolio. In columns (3) and (4) we report the *Cumulative Abnormal Sentiment Activity* for both event

**Table 3. Stock market reactions and sentiment reactions to government quarantine announcements.**

| | IGPA CAR (Market Model SP500) | | Twitter CASA | | Google Trend CASVA | |
|---|---|---|---|---|---|---|
| | (1) | (2) | (3) | (4) | (5) | (6) |
| Announcement | (-1,0) | (-1,+1) | (-1,0) | (-1,+1) | (-1,0) | (-1,+1) |
| 24–03–2020 | -7,35% *** | -1,39% | 14,49% ** | 7,05% | -17,06% *** | -27,43% *** |
| 27–03–2020 | 2,11% ** | 3,58% *** | -6,60% *** | -3,51% | -35,20% *** | -53,56% *** |
| 30–03–2020 | 2,98% *** | 8,58% ** | -1,33% | -19,33% ** | -40,85% *** | -43,76% ** |
| 31–03–2020 | 7,07% ** | 6,84% ** | -14,90% ** | -31,08% ** | -21,27% ** | -30,32% ** |
| 02–04–2020 | 2,73% ** | 5,99% ** | -17,59% ** | -19,06% ** | -16,46% *** | -5,46% |
| 06–04–2020 | 2,71% ** | 2,04% | -11,81% ** | -12,43% ** | 16,70% *** | 31,37% *** |
| 07–04–2020 | -1,23% *** | -0,87% ** | -10,96% ** | 1,90% | 20,36% ** | 27,20% *** |
| 14–04–2020 | 1,69% ** | 0,98% | 1,43% ** | -2,61% | 3,53% | 25,11% ** |
| 21–04–2020 | -2,36% *** | -2,53% ** | -13,96% ** | -23,90% ** | 6,60% ** | -0,47% |
| 28–04–2020 | 4,90% *** | 6,72% *** | 46,35% *** | 64,79% *** | -13,05% *** | 9,51% |
| 04–05–2020 | -3,27% ** | -1,92% | 6,98% | -24,90% | 18,08% | 14,88% |
| 06–05–2020 | 2,60% *** | 3,01% *** | -41,07% ** | -23,67% | 25,40% ** | 23,46% ** |
| 13–05–2020 | -3,85% ** | -3,43% ** | 3,39% | -640,94% ** | 32,05% *** | 50,69% *** |
| 20–05–2020 | -0,03% | -0,82% ** | -21,22% ** | -4,91% | 16,03% ** | 22,83% ** |
| 27–05–2020 | -0,64% | -3,09% ** | 0,27% | 22,26% ** | -22,09% ** | -23,15% ** |
| 08–06–2020 | 4,85% *** | 4,85% ** | 1,45% | -12,70% | 8,70% | 6,84% |
| 10–06–2020 | -1,21% ** | -1,81% ** | -17,87% ** | -13,58% ** | 8,43% ** | -2,79% |
| 17–06–2020 | 1,91% ** | 1,81% ** | -22,68% *** | -22,49% ** | 31,85% ** | 1,56% |
| 19–06–2020 | 1,07% ** | 0,33% | 3,73% ** | 8,17% ** | -60,94% *** | -28,28% |
| 22–06–2020 | 0,43% | 1,03% | 7,98% *** | -0,51% | 2,01% | 33,36% |
| 24–06–2020 | 1,62% *** | 1,31% ** | -10,73% ** | -6,49% | 3,89% | -6,96% |
| 08–07–2020 | -1,99% ** | -5,24% ** | 13,95% ** | 18,50% *** | -21,26% ** | -55,55% ** |
| 09–07–2020 | -5,10% *** | -5,58% ** | 15,11% ** | 14,95% ** | -55,02% *** | -69,13% *** |
| 13–07–2020 | 1,90% | 2,82% ** | -19,31% ** | -33,90% ** | 22,47% | 50,60% ** |
| 20–07–2020 | -3,21% *** | -4,12% *** | -1,98% ** | 8,07% ** | 3,52% | 7,55% |

Note:

$^{+}$p<0.15;

$^{*}$p<0.1,

$^{**}$p<0.05;

$^{***}$p<0.01.

windows, and in columns (5) and (6) we present the observed *Cumulative Abnormal Search Volume Activity* upon lockdown announcements.

For both event windows and for any of the stock or market sentiment variables considered, we observe positive and negative abnormal reactions. This situation is most likely to arise from the fact that each lockdown announcement involves some municipalities going into lockdown and others going out of it. We hypothesize that the observed heterogeneity in stock market and sentiment reactions reflects the number of people going into lockdown and their *SES*.

In Fig 5 we present a *heat map* for the correlation between the *IGPA* index, the *Sentiment Score* and the *Average Search Volume Index—ASVI-* series for our sample period. It is possible to observe a strong and negative correlation between the levels of the *IGPA* and the *ASVI* series, which suggests that during periods of high stock market valuation, markets concerns about the development of the pandemic are dimmed. Concerning the correlations between the

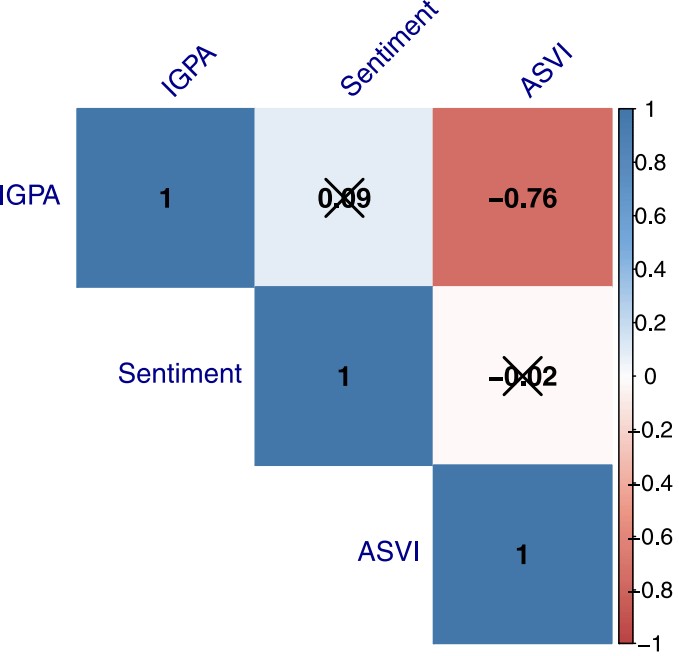

**Fig 5. Overview of correlations visualized in a correlation matrix for the IGPA index, sentiment score and ASVI.** *p* —value was set at 0.1 and x marks all bivariate correlations that were not significant.

stock index and the *sentiment score* and between this last variable and the *ASVI*, correlations are very low and statistically insignificant at any standard level of significance.

In Figs 6 and 7, we present *heat maps* for the correlation between the *IGPA CAR*, the *CASA* and the *CASVA* reported in Table 3. For both event windows, we observe significant and

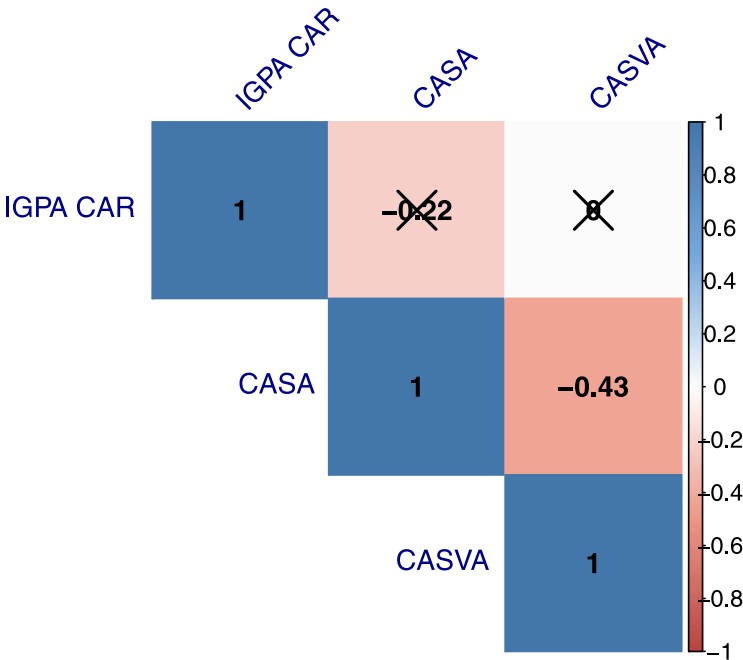

**Fig 6. Overview of correlations visualized in a correlation matrix for the (−1, 0).** *p*—value was set at 0.1 and x marks all bivariate correlations that were not significant.

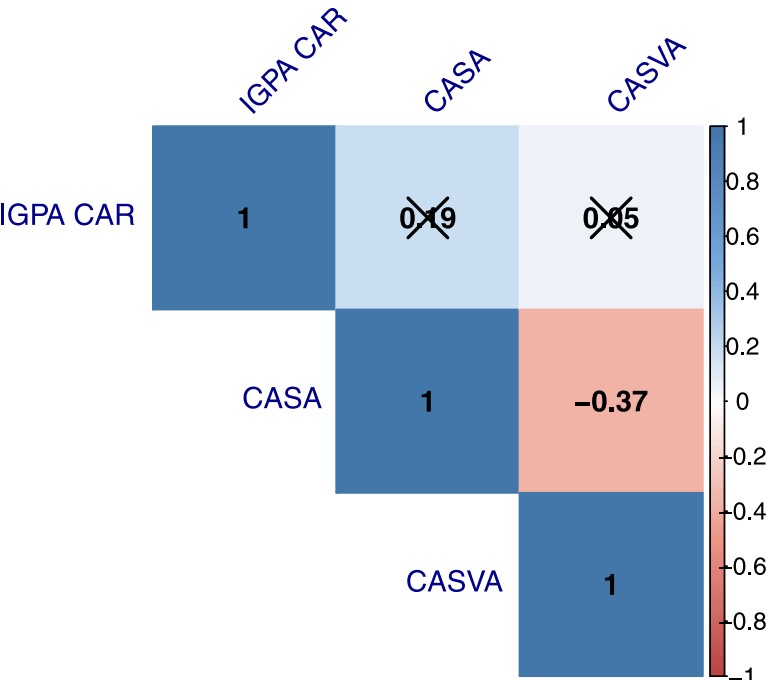

**Fig 7. Overview of correlations visualized in a correlation matrix for the (−1, +1).** *p*—value was set at 0.1 and x marks all bivariate correlations that were not significant.

negative correlations between our sentiment proxies. As expected, since these measures tend to move in opposite directions in response to good or bad news, an increase (decrease) in the abnormal search activity is associated with a more pessimistic (optimistic) sentiment during the windows centered around lockdown announcements. The correlation between the abnormal stock returns and the sentiment proxies is low and statistically insignificant for either window. Interestingly, this result is consistent with the results in Kim et al [25]. These authors show that Google searches are not correlated with contemporary stock returns, nor can they predict future abnormal returns.

### 4.2 Stock market reactions and *SES*

Regarding stock market reactions to lockdown announcements, we estimate Eq 5 for each of the government announcements in Table 3. For discussion and analysis, we consider particularly relevant the population belonging to the *ABC1* segment when comparing the results between the wealthiest and total population. In any case, to show that our results are not driven by the selection of an arbitrary cohort of the population, we consider six cohorts based on the *MPI* sorting for the change in confined population ($\Delta Population_{i,t}$). Results are presented in panel A of Table 4. The first cohort is the population belonging to the five municipalities featuring the lowest *MPI*, i.e., the wealthiest municipalities of the country according to this sorting, comprising 4.68% of the population. The second cohort corresponds to the population belonging to the top ten municipalities according to the *MPI* sorting, with 9.57% of the population. The rest of the cohorts are defined likewise, except the last one, that corresponds to the whole population. The third cohort comprises 12.76% of the richest population, a figure that roughly corresponds to the *ABC*1 socioeconomic segment of the country.

Each column in Table 4 presents the results for the estimation of Eq 5 for a specific cohort. The first six columns of the table report the results for the (−1, 0) window; the last six columns

**Table 4. Market returns vs. changes in quarantined population (Population in millions).**

| | IGPA Cumulative Abnormal Return | | | | | | | | | | | |
|---|---|---|---|---|---|---|---|---|---|---|---|---|
| | Panel A: Market Model (S&P 500)—Municipalities sorted according to the Multidimensional Poverty Index | | | | | | | | | | | |
| | (-1,0) | | | | | | (-1,+1) | | | | | |
| | (1) | (2) | (3) | (4) | (5) | (6) | (7) | (8) | (9) | (10) | (11) | (12) |
| Δ Wealthiest Population | | | | | | | | | | | | |
| Wealthiest 5 Mun. (4.68%) | -0.0766*** | | | | | | -0.0428*** | | | | | |
| | (0.0221) | | | | | | (0.0104) | | | | | |
| Wealthiest 10 Mun. (9.57%) | | -0.0462** | | | | | | -0.0266*** | | | | |
| | | (0.0219) | | | | | | (0.0058) | | | | |
| Wealthiest 15 Mun. (12.76%) (*ABC*1) | | | -0.0447* | | | | | | -0.0264*** | | | |
| | | | (0.0239) | | | | | | (0.0051) | | | |
| Wealthiest 20 Mun. (18.20%) | | | | -0.0395** | | | | | | -0.0214*** | | |
| | | | | (0.0170) | | | | | | (0.0059) | | |
| Wealthiest 24 Mun. (22.37%) | | | | | -0.0336** | | | | | | -0.0159 + | |
| | | | | | (0.0139) | | | | | | (0.0104) | |
| Δ Total Population | | | | | | -0.0136 | | | | | | -0.0009 |
| | | | | | | (0.0135) | | | | | | (0.0154) |
| Observations | 25 | 25 | 25 | 25 | 25 | 25 | 25 | 25 | 25 | 25 | 25 | 25 |
| R2 | 0.29 | 0.24 | 0.24 | 0.26 | 0.19 | 0.05 | 0.07 | 0.06 | 0.06 | 0.06 | 0.03 | 0.00 |
| | Panel B: Market Model (S&P 500)—Municipalities sorted according to Municipal Income | | | | | | | | | | | |
| | (-1,0) | | | | | | (-1,+1) | | | | | |
| | (1) | (2) | (3) | (4) | (5) | (6) | (7) | (8) | (9) | (10) | (11) | (12) |
| Δ Wealthiest Population | | | | | | | | | | | | |
| Wealthiest 5 Mun. (7.18%) | -0.0662** | | | | | | -0.024 | | | | | |
| | (0.0260) | | | | | | (0.0180) | | | | | |
| Wealthiest 10 Mun. (15.30%) | | -0.0377** | | | | | | -0.0191* | | | | |
| | | (0.0135) | | | | | | (0.0097) | | | | |
| Wealthiest 15 Mun. (22.54%) | | | -0.0332** | | | | | | -0.0162 | | | |
| | | | (0.0143) | | | | | | (0.0126) | | | |
| Wealthiest 20 Mun. (28.26%) | | | | -0.0286** | | | | | | -0.0149 | | |
| | | | | (0.0123) | | | | | | (0.0112) | | |
| Wealthiest 24 Mun. (32.01%) | | | | | -0.0268** | | | | | | -0.0145 | |
| | | | | | (0.0116) | | | | | | (0.0102) | |
| Δ Total Population | | | | | | -0.0136 | | | | | | -0.0009 |
| | | | | | | (0.0135) | | | | | | (0.0154) |
| Observations | 25 | 25 | 25 | 25 | 25 | 25 | 25 | 25 | 25 | 25 | 25 | 25 |
| R2 | 0.21 | 0.24 | 0.19 | 0.18 | 0.18 | 0.05 | 0.02 | 0.05 | 0.03 | 0.04 | 0.04 | 0.00 |

Note:

+ p<0.15;

* p<0.1;

** p<0.05;

*** p<0.01

show the results obtained for the (−1, +1) window. In the analysis that follows, we resort to the Cribari-Neto *HC*4 heteroskedasticity-consistent covariance matrix estimators for making inferences. Cribari-Neto [51] shows that this estimator performs well in small samples, especially in the presence of influential observations.

For both event windows, we obtain a monotonically decreasing magnitude of market reactions upon announcements as we move from the wealthiest municipalities to the whole population. For the (−1, 0) window in column (1), an increase of one million people in the wealthiest segment of the population under lockdown produces a more negative *CAR* by close to 800 basis points. This effect is statistically significant and economically meaningful. In column (3), an increase of one million people in the *ABC1* segment under lockdown, produces a more negative *CAR* in the (−1, 0) window by close to 450 basis points, significant at the 10% level and achieving an $R^2$ coefficient near to 0.24. When we consider the population belonging to the municipalities in the highest wealth quintile in column (5), i.e., the first 24 wealthiest municipalities out of the 120 considered in our sample, the effect on *CARs* drops to nearly half of what we obtained in column (1), but remains both economically and statistically significant at a 5% level, with an $R^2$ close to 0.2. For the changes in the total population under lockdown in column (6), the effect on stock market reactions declines even further and becomes statistically insignificant. Furthermore, $R^2$ declines drastically.

For the (−1, +1) window in columns (7) to (12), the market reactions are smaller in magnitude than for the (−1, 0) window, with lower $R^2$ coefficients. However, estimates remain significantly different from zero for the wealthiest municipalities. Again, the effect vanishes when changes in total population are considered. The higher fits and the higher point estimates observed for the (−1, 0) are consistent with the timing of the announcements, which are typically made before noon, and with market reactions taking place on that same day. In any case, the monotonically decreasing market reactions upon government announcements for the (−1, +1) window are still observed as we move from the wealthiest municipalities to the whole population.

The results presented in panel A of Table 4 are obtained for a sorting of the population based on the *MPI* of the corresponding municipalities. As a robustness check, we consider an alternative sorting, based on *Municipal Income*, available at https://observatoriofiscal.cl/Informate/Repo/BrechasentreMunicipios. Municipal income includes all sources of financing available to municipalities; collection of business licenses, income from land and property taxes, payment of road taxes, fines collected by the municipality, as well as transfers from the Central Government. However, it does not include factors related to health, education, and living standards that the *MPI*, our preferred sorting variable, does include. Results are presented in panel B of Table 4. For the (−1, 0) window, we obtain similar results, both in magnitude and statistical significance, to those obtained for the *MPI* sorting. Even though the goodness-of-fit of the regressions is somewhat dimmed, the *R2* coefficients remain high and the monotonically decreasing market reactions are still observed as we move from high to low-income municipalities. For the (−1, +1) window, point estimates are smaller than those presented in panel A and exhibit low statistical significance.

In sum, the results in Table 4 provide evidence that stock market reactions to lockdown announcements depend on the *SES* of the population under lockdown. Recognizing that there might be several reasons why such a phenomenon could be observed, it strongly suggests a high level of wealth concentration among the richer population. In fact, according to the World Inequality Database (https://wid.world/), as of year-end 2018, the top 10% wealthiest population accounts for 60.4% of the total income of the country. As richer cohorts are considered for the changes in the number of people under lockdown, the predicted abnormal returns in the stock market are higher in magnitude. Furthermore, changes in the total population cannot explain stock market reactions to such announcements. This result is significant because it validates our proposed wealth ranking, which will also be used to analyze sentiment responses to government announcements below.

## 4.3 Sentiment responses and *SES*

Just as in the case of the cumulative abnormal returns for the stock market, we observe positive and negative cumulative abnormal responses to our sentiment proxies. To analyze whether sentiment reactions depend on the number and the socioeconomic characteristics of the people under lockdown, we estimate regression equations Eqs (14) and (15), which relate our abnormal sentiment measures to the government announcements in Table 3 to the number of people under lockdown. In either equation, just as in the case of stock returns, $\Delta Population_{i,t}$ refers to changes in the number of people from cohort "$i$" under lockdown at the announcement made at time $t$. For the controls in $x_t$, we consider the cumulative abnormal returns of the IGPA index for the corresponding event window, and the prevalent value of the *Stringency Index* at the day of the announcement. The stock market's abnormal returns are included to control for the possible effect that stock market performance might have on market-wide sentiment [20]. We include the *Stringency Index* to control for the effect that policy responses to the pandemic might have on market sentiment and can be considered a proxy for the severity of the disease. This index, developed in [52] and available at [53], ranges from 1 to 100 and records the strictness of lockdown style policies implemented by governments around the globe. It has been widely used in the recent literature on the economic and social effects of the COVID-19 pandemic [54, 55].

Estimation results for the *Cumulative Abnormal Sentiment Activity* in Eq (14) are presented in Table 5. In columns (1) to (6) we present results for the (−1, 0) event window. In columns (7) to (12) we report the results for the (−1, +1) window. It should be noted that the cumulative abnormal return of the IGPA index, included as a control in all specifications, is a generated regressors. As such, the variability from the *first stage* estimation of the control should be considered when performing inferences for the estimated parameters of equation Eq (14) [56]. To make sure our results are valid, we compute and report statistical significance using *nonparametric bootstrapped standard errors* alongside the significance obtained by means of the Cribari-Neto *HC*4 robust standard error estimator, as explained in *Note 2* in Table 5[57].

For the (−1, +1) windows, we obtain negative and statistically significant coefficients for the change in the wealthiest cohorts of the population under lockdown. For the *ABC1* socioeconomic segment in column (9), we obtain a point estimate of −3.29, significant at the 10% (5%) level when Cribari-Neto *HC*4 (bootstrapped) standard errors are used. This is a very large economic effect. An increase of one million people under lockdown in this segment produces a negative abnormal sentiment reaction close to 330% percent. Notably, the goodness-of-fit of these specifications is 73%.

In columns (12), the effect of total population changes under lockdown on abnormal sentiment is also negative, but the point estimate of −1.72 is nearly half of the one obtained for the *ABC1* segment. It is still significant at the 15% or 10% level, depending on the estimator used to compute standard errors, and the specification features a rather large $R^2$ coefficient close to 0.55. Since we are interested in assessing whether market sentiment responds differently to different socioeconomic cohorts of the population under lockdown, we perform a *Wald* test for the equality of the population variable coefficient between specifications (9) and (12). The (unreported) test rejects the null hypothesis of equality of coefficients at any standard level of significance, using either robust *HC*4 or bootstrapped errors for the covariance matrix.

Similarly to the phenomenon observed for the abnormal returns of the stock market, we obtain a nearly monotonically decreasing magnitude of market sentiment responses to government announcements as we move from the wealthiest municipalities in column (7) to the whole population in column (12). For instance, for the richest 4.68% of the population in column (7), the point estimate of the change in population variable reaches a value of −4.4, two

**Table 5. Sentiment (Twitter) vs. changes in confined population (Population in millions).**

| | Twitter | | | | | | | | | | | |
|---|---|---|---|---|---|---|---|---|---|---|---|---|
| | Cumulative Abnormal Sentiment Activity—*MPI* Sorting | | | | | | | | | | | |
| | (-1,0) | | | | | | (-1,+1) | | | | | |
| | **(1)** | **(2)** | **(3)** | **(4)** | **(5)** | **(6)** | **(7)** | **(8)** | **(9)** | **(10)** | **(11)** | **(12)** |
| Municipalities | | | | | | | | | | | | |
| Δ Top 5 (4.68%) | 0.1816 | | | | | | $-4.3917^{(,+)}$ | | | | | |
| | (0.1758) | | | | | | (3.2453) | | | | | |
| Δ Top 10 (9.57%) | | 0.1087 | | | | | | $-3.2348^{(*,*)}$ | | | | |
| | | (0.1370) | | | | | | (1.8239) | | | | |
| Δ Top 15 (12.76%) (*ABC*1) | | | 0.0781 | | | | | | $-3.2864^{(*,**)}$ | | | |
| | | | (0.1073%) | | | | | | (1.6531) | | | |
| Δ Top 20 (18.20%) | | | | 0.0460 | | | | | | $-2.8497^{(*,*)}$ | | |
| | | | | (0.0561) | | | | | | (1.5510) | | |
| Δ Top 24 (22.37%) | | | | | 0.0068 | | | | | | $-2.6648^{(*,*)}$ | |
| | | | | | (0.0681) | | | | | | (1.5462) | |
| Δ Total Pop. | | | | | | −0.0257 | | | | | | $-1.7160^{(+,*)}$ |
| | | | | | | (0.0716) | | | | | | (1.1257) |
| Controls | | | | | | | | | | | | |
| IGPA CAR | −0.4305 | −0.5581 | −0.7018 | −0.8147 | −1.0683 | −1.1995 | −4.0296 | −3.7241 | −4.4542 | −5.3600 | −2.9511 | 3.4379 |
| | (1.4895) | (1.4181) | (1.4075) | (1.4132) | (1.4762) | (1.2782) | (5.5674) | (5.1629) | (5.0076) | (5.5422) | (5.1793) | (5.8877) |
| Stringency Index | 0.0012 | 0.0006 | 0.0002 | 0.0002 | −0.0006 | −0.0015 | −0.0544 | −0.0464 | −0.0502 | −0.0683 | −0.0636 | −0.0411 |
| | (0.0044) | (0.0045) | (0.0044) | (0.0042) | (0.0040) | (0.0046) | (0.0729) | (0.0598) | (0.0561) | (0.0671) | (0.0644) | (0.0471) |
| Observations | 25 | 25 | 25 | 25 | 25 | 25 | 25 | 25 | 25 | 25 | 25 | 25 |
| $R^2$ | 0.0867 | 0.0846 | 0.0686 | 0.0575 | 0.0494 | 0.0553 | 0.5080 | 0.6774 | 0.7322 | 0.6948 | 0.6521 | 0.5337 |

Note 1: Standard errors in parenthesis computed using Cribari-Neto $HC$4 robust estimator.

Note 2: Significance levels in (.,.). First and second entry corresponds to $HC$4 and Bootstrapped estimators, respectively.

[+]$p < 0.15$;

[*]$p < 0.1$,

[**]$p < 0.05$;

[***]$p < 0.01$

and half times bigger than the coefficient obtained for the total population in column (12), though it is only significant at the 15% level using bootstrapped errors. In column (11), when we consider the municipalities from the first wealth quintile that comprises 22.37% of the country's population, the point estimate drops to −2.77, but remains both economically and statistically significant at the 10% level, with an $R^2$ close to 0.65.

For the (−1, 0) window, all specifications return statistically insignificant coefficients, regardless of the type of standard errors used for inference, and exhibit lower fits than for the (−1, +1) window.

The results in Table 5 constitute novel evidence on the *SES* of Twitter users. The observed relationship between *Abnormal Sentiment Activity* and changes in the characteristics of people under lockdown suggest a socioeconomic segregation among the users of the platform. Our results are in line with the results of a survey carried out by the *Pew Research Center* in the US, in which Twitter users appear to be more highly educated and having higher incomes than the rest of the population.

**Table 6. Sentiment (GT) vs. changes in confined population (Population in millions).**

| | Google Trends | | | | | | | | | | | |
|---|---|---|---|---|---|---|---|---|---|---|---|---|
| | Cumulative Abnormal Search Volume Activity—*MPI Sorting* | | | | | | | | | | | |
| | (-1,0) | | | | | | (-1,+1) | | | | | |
| | **(1)** | **(2)** | **(3)** | **(4)** | **(5)** | **(6)** | **(7)** | **(8)** | **(9)** | **(10)** | **(11)** | **(12)** |
| Municipalities | | | | | | | | | | | | |
| Δ Top 5 (4.68%) | 0.0586 | | | | | | 0.0439 | | | | | |
| | (0.4306) | | | | | | (0.6057) | | | | | |
| Δ Top 10 (9.57%) | | 0.0679 | | | | | | 0.0932 | | | | |
| | | (0.3281) | | | | | | (0.4415) | | | | |
| Δ Top 15 (12.76%) (*ABC*1) | | | 0.1054 | | | | | | 0.1350 | | | |
| | | | (0.2982) | | | | | | (0.4173) | | | |
| Δ Top 20 (18.20%) | | | | 0.1105 | | | | | | 0.1229 | | |
| | | | | (0.2275) | | | | | | (0.3420) | | |
| Δ Top 24(22.37%) | | | | | 0.1472 | | | | | | 0.1542 | |
| | | | | | (0.1518) | | | | | | (0.2699) | |
| Δ Total Pop. | | | | | | 0.1532$^{(**,*)}$ | | | | | | 0.1441 |
| | | | | | | (0.0636) | | | | | | (0.1249) |
| Controls | | | | | | | | | | | | |
| IGPA CAR | 0.2121 | 0.3362 | 0.5390 | 0.6918 | 0.8010 | 0.5566 | 0.4867 | 0.6855 | 0.8471 | 0.9097 | 0.9527 | 0.6763 |
| | (2.2030) | (2.1866) | (2.1907) | (2.2184) | (2.0242) | (1.7329) | (2.3587) | (2.3836) | (2.4206) | (2.4347) | (2.2983) | (2.1551) |
| Stringency Index | −0.0005 | −0.0003 | 0.0003 | 0.0012 | 0.0022 | 0.0033 | −0.0003 | 0.0008 | 0.0017 | 0.0026 | 0.0035 | 0.0037 |
| | (0.0098) | (0.0095) | (0.0093) | (0.0097) | (0.0093) | (0.0086) | (0.0164) | (0.0156) | (0.0156) | (0.0169) | (0.0162) | (0.0144) |
| Observations | 25 | 25 | 25 | 25 | 25 | 25 | 25 | 25 | 25 | 25 | 25 | 25 |
| $R^2$ | 0.0028 | 0.0070 | 0.0164 | 0.0218 | 0.0423 | 0.0946 | 0.0039 | 0.0113 | 0.0212 | 0.0220 | 0.0349 | 0.0570 |

Note 1: Standard errors in parenthesis computed using Cribari-Neto *HC*4 robust estimator.

Note 2: Significance levels in (.,.). First and second entry corresponds to *HC*4 and Bootstrapped estimators, respectively.

$^+$p<0.15;

$^*$p<0.1,

$^{**}$p<0.05;

$^{***}$p<0.01

The estimation results of Eq 15 for the *Cumulative Abnormal Search Volume Activity* of Google Trends are presented in Table 6. For the controls in $x_t$, we consider again the cumulative abnormal returns of the IGPA index for the corresponding event window, and the prevalent value of the *Stringency Index* at the day of the announcement. Since the *CARs* of the IGPA index are generated regressors, we report significance both for bootstrapped standard errors and *HC*4 robust standard errors, as explained in *Note 2* of the table.

For the (−1, 0) event window, when we consider changes in the total population under lockdown, we obtain a positive effect of those changes on abnormal search activities. In column (6), an increase of one million people in the total population under lockdown increases the abnormal volume of search activity by nearly 15%, a rather large effect considering the magnitudes of the reactions presented in column (5) of Table 3. For the rest of the cohorts in columns (1) to (5), the estimated coefficients are statistically insignificant, with much smaller $R^2$ coefficients. For the (−1, +1) window in columns (7) to (12), our point estimators are similar to the ones we obtained for the (−1, 0) window. Still, they all turn out to be statistically insignificant.

Based on the evidence presented in Table 6, among the users of Google queries, there seems to be no socioeconomic segregation, as measured by a truncation of municipalities based on the *MPI*. When the total number of people under lockdown increases, an abnormal increase in the pandemic-related Google searches is observed. Such an effect is smaller in magnitude and statistically insignificant when changes in the wealthiest populationidered. In any case, this evidence should be taken with caution. Google Trends counts aggregate "searches" and not the people who perform them. A priori, it does not reveal whether a spike in the relative proliferation of a search term is due to a few power users or many infrequent users. Finally, there is some evidence that abnormal search activity appears to be concentrated in the shorter (-1,0) window, a phenomenon that suggests instantaneity and the short life of Google queries in pandemic-related news.

As an alternative approach to analyze the impact of lockdown announcements on market sentiment, we consider a panel data regression similar to the *Difference in Difference (DiD)* methodology, which allows us to take advantage of the panel structure of our data, increasing considerably the sample size for the estimation. It should be noted, however, that the sample size achieved is still rather small, and results should be interpreted in light of this limitation. As explained in the Methods section, we estimate the specification presented in Eq (17) for the *Abnormal Sentiment Activity* index and the *Abnormal Search Volume Activity* index. To make results comparable to our previous results, we consider the three days in the $(-1,1)$ window centered around each announcement day. For this window, equation Eq (17) can be written as:

$$
\begin{aligned}
y_t = {} & \alpha_1 + \delta_{-1} D_{it}^{(-1)} + \delta_0 D_{it}^{(0)} + \delta_{+1} D_{it}^{(+1)} + \\
& \beta_{-1} D_{it}^{(-1)} W_i + \beta_0 D_{it}^{(0)} W_i + \beta_{+1} D_{it}^{(+1)} W_i + \gamma W_i + \mu_{it}
\end{aligned}
\tag{18}
$$

To see what effects the parameters in Eq (18) capture, let's assume that we are interested in the sentiment responses on the day after government announcements are made. It is straightforward to see that the expected difference in sentiment between locked down and not locked down wealthy municipalities is given by $\delta_{+1} + \beta_{+1}$. Also, the expected difference in sentiment between locked down and not locked down, non-wealthy municipalities is $\delta_{+1}$. The parameter $\beta_{+1}$ is then analogous to the standard difference-in-difference estimator; i.e., it is the average difference in the expected outcome between confined and not confined wealthy municipalities and between locked down and not locked down non-wealthy municipalities, where the average is taken for all the days in the sample that correspond to the day after the government makes an announcement.

Results for the *ASA* index are presented in columns (1) to (4) of Table 7 and in Fig 8. Results for the *ASVA* index are reported in columns (5) to (8) of the same table. Specifications differ in the inclusion of controls. We consider the abnormal return, *AR*, of the *IGPA* index, and the value of the *Stringency Index* as control variables. Given our sample size, we present standard errors computed using *White HC0* robust standard errors. Since the abnormal return of the IGPA index is a generated regressor, we report significance both for bootstrapped standard errors and *HC0* robust standard errors, as explained in *Note 2* of the table. For estimation, the 15 municipalities with the lowest *MPI* are considered wealthy. They correspond to the *ABC1* socioeconomic segment that comprises 12.76% of the total population.

For the *ASA* index, the results in panel A of Table 7 show that the largest difference in social sentiment responses between wealthy and non-wealthy municipalities occurs the day after government announcements are made. In column (1), when no controls are included, the parameter estimate for $\beta_{+1}$ is negative and statistically significant at the 1% level. The effect is also economically meaningful. The expected abnormal sentiment response upon lockdown is

**Table 7. Panel estimation for sentiment proxies—$ASA_t$ and $ASVA_t$.**

| | Abnormal Sentiment Activity—ASA | | | | Abnormal Search Volume Activity—ASVA | | | |
|---|---|---|---|---|---|---|---|---|
| | (1) | (2) | (3) | (4) | (5) | (6) | (7) | (8) |
| Panel A: Difference in Difference "Like" Estimator | | | | | | | | |
| Day before ($\beta_{-1}$) | -0.0531$^{(+,+)}$ | -0.0470 | -0.0698$^{(*,*)}$ | -0.0632$^{(+,+)}$ | 0.0191 | 0.0443 | 0.0423 | 0.0655$^{(+,+)}$ |
| | (0.0344) | (0.0336) | (0.0370) | (0.0486) | (0.0525) | (0.0544) | (0.0510) | (0.0405) |
| Announcement day ($\beta_0$) | -0.0896$^{(**,**)}$ | -0.0785$^{(**,**)}$ | -0.0436 | -0.0303 | 0.0472 | 0.0927$^{(*,+)}$ | -0.0166 | 0.0299 |
| | (0.0376) | (0.0376) | (0.0398) | (0.0411) | (0.0662) | (0.0553) | (0.0708) | (0.0547) |
| Day after ($\beta_{+1}$) | -2.3578$^{(***,***)}$ | -2.3448$^{(***,***)}$ | -2.3146$^{(***,***)}$ | -2.2992$^{(***,***)}$ | 0.0711 | 0.1244$^{(**,**)}$ | 0.1090 | 0.0649 |
| | (0.8392) | (0.8415) | (0.8393) | (0.8417) | (0.0601) | (0.0601) | (0.0647) | (0.0575) |
| Panel B: Parameter Estimates | | | | | | | | |
| $\gamma$ | 0.08124$^{(***,***)}$ | 0.0693$^{(**,**)}$ | 0.0344 | 0.2021 | -0.0713$^{(*,*)}$ | -0.1203$^{(***,***)}$ | 0.0064 | -0.0562$^{(+,)}$ |
| | (0.0273) | (0.0288) | (0.0292) | (0.0329) | (0.0411) | (0.0408) | (0.0510) | (0.0383) |
| $\delta_{-1}$ | 0.0271 | 0.0204 | 0.0702$^{(**,**)}$ | 0.0633$^{(**,*)}$ | 0.1306$^{(***,***)}$ | 0.1025$^{(***,***)}$ | 0.0709$^{(**,**)}$ | 0.0466$^{(*,*)}$ |
| | (0.2153) | (0.0239) | (0.0298) | (0.0319) | (0.0290) | (0.0263) | (0.0299) | (0.0268) |
| $\delta_0$ | 0.0569$^{(**,**)}$ | 0.0617$^{(**,**)}$ | 0.0586$^{(**,**)}$ | 0.0641$^{(**,*)}$ | 0.1017$^{(***,***)}$ | 0.1216$^{(***,***)}$ | 0.0994$^{(***,**)}$ | 0.1186$^{(***,***)}$ |
| | (0.0255) | (0.0262) | (0.0259) | (0.0270) | (0.0369) | (0.0346) | (0.0381) | (0.0354) |
| $\delta_{+1}$ | -0.6854$^{(**,***)}$ | -0.6830$^{(**,**)}$ | -0.6817$^{(**,*)}$ | -0.6789$^{(**,)}$ | 0.0236 | 0.0334 | 0.0184 | 0.0281 |
| | (0.2812) | (0.2814) | (0.2811) | (0.2812) | (0.0341) | (0.0335) | (0.0356) | (0.0341) |
| Panel C: Controls | | | | | | | | |
| IGPA AR | | -0.7963 | | -0.8952 | | -3.2670$^{(***,***)}$ | | -3.1405$^{(***,***)}$ |
| | | (0.7521) | | (0.7530) | | (0.3232) | | (0.2795) |
| Stringency Index | | | -0.0073$^{(***,***)}$ | -0.0070$^{(**,***)}$ | | | 0.0099$^{(***,***)}$ | 0.0095$^{(***,***)}$ |
| | | | (0.0025) | (0.0025) | | | (0.0010) | (0.0008) |
| Observations | 325 | 325 | 325 | 325 | 325 | 325 | 325 | 325 |
| R2 | 0.2632 | 0.2634 | 0.2643 | 0.2645 | 0.1203 | 0.2617 | 0.2527 | 0.3829 |

Note 1: Standard errors in parenthesis computed using White $HC0$ robust estimator.

Note 2: Significance levels in (.,.). First and second entry corresponds to $HC0$ and Bootstrapped estimators, respectively.

$^+$p<0.15;

$^*$p<0.1,

$^{**}$p<0.05;

$^{***}$p<0.01

more than 4 times more negative for wealthy municipalities than for non-wealthy municipalities ($\frac{\beta_{+1}+\delta_{+1}}{\delta_{+1}} = \frac{-2.36-0.69}{-0.69} = 4.4$). This result is almost identical for all specifications, regardless of the inclusion of controls.

For the days preceding announcement days, the estimator $\beta_{-1}$ is relatively small and it is statistically insignificant for all specifications at the 10% level, except for the estimation in column (3). For the announcement day, the $\beta_0$ is negative and statistically significant in columns (1) and (2), but loses significance whenever the *Stringency Index* is included in specifications (3) and (4). In fact, when both controls are included in the later specification, only the estimate for $\beta_{+1}$ is statistically significant at the 10% level. These results are in line with those presented in Table 5 where we document that, controlling for stock market performance and the level of the *Stringency Index*, changes in the number of people from the wealthiest municipalities under lockdown have explanatory power over the cumulative abnormal sentiment activity variable, *CASA*, but only for the (−1, +1) window.

Regarding the controls, the *Stringency Index* estimate is negative and significant, which suggests that the strictness of lockdown policies implemented by the government, or the severity

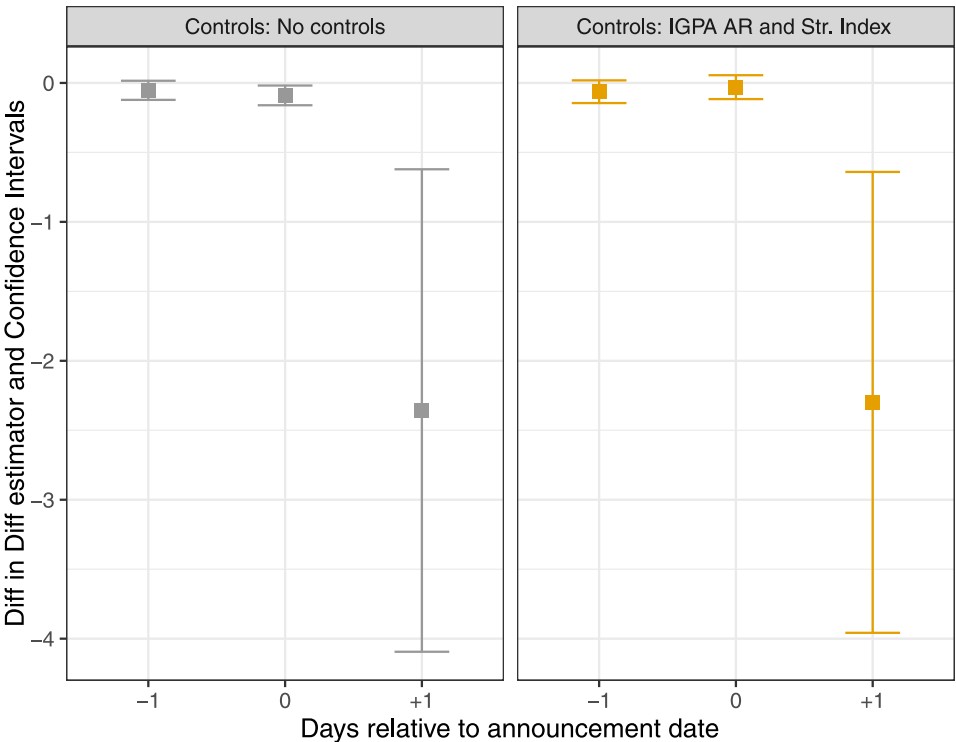

**Fig 8. *ASA*—Difference-in-difference "like" estimators ($\beta_{-1}$, $\beta_0$, $\beta_{+1}$).** Average difference in the expected *Abnormal Sentiment Activity* index between locked down and not locked down wealthy municipalities and between locked down and not locked down non-wealthy municipalities.

of the pandemic to which lockdown policies respond, has a negative impact on overall market sentiment. The stock market's abnormal performance, proxied by the *IGPA AR*, is not significant in our estimations.

For the *ASVA* index, the results in Table 7 provide no evidence of a statistically significant difference in the abnormal volume of pandemic-related searches in response to government announcements between wealthy and non-wealthy municipalities. All the estimators in Panel A turn out to be statistically insignificant, with the exception of those corresponding to specification (6). In any case, whenever the *Stringency Index* is included as a control, all estimated coefficients result statistically insignificant at the 10% level. Interestingly, the $\delta$ parameters in Panel B, that in this case capture the effects of quarantine announcements on the *ASVA* index for non-wealthy municipalities, are positive and highly significant for the same day and the day preceding government announcements, but not for the day after them. These results seem to reflect the high levels of anxiety regarding potential lockdown measures that government announcements produce on the population. Furthermore, the results presented in Table 7 are consistent with our previous results presented in Table 6, in which increases in the total population under lockdown increases the abnormal volume of search activity, but with statistical significance only for the $(-1, 0)$ window.

The controls included in the empirical specification have the expected signs and are highly significant across the different specifications. The *Stringency Index* exhibits a positive sign, which suggests that the prevalence of stricter lockdown policies produces more pandemic-related internet searches. For the performance of the stock market, we observe that the higher the abnormal returns of the stock market surrounding announcements, the lower the volume

of pandemic-related queries. This is in line with the correlations presented in Fig 5, where we document a strong and negative correlation between the levels of the *IGPA* and the *ASVI* series, which suggests that during periods of high stock market valuation, market participants care less about the development of the pandemic. Even though recent financial literature shows that causality might run in the opposite direction, with Google search volumes being able to predict stock returns [18, 25, 58], we acknowledge that in our case it is hard to claim that causality actually runs in that direction. We consider a short time span around an impactful event, the government announcement, that is likely to affect both the stock market and Google search volumes at the same time.

## 5 Conclusion

The Chilean health authorities' strategic quarantine scheme provides a unique opportunity to assess the impact of lockdowns on social sentiment. Whereas generalized lockdowns affect the population of a country as a whole, dynamic or strategic quarantines affect different parts of the population at different times. The high level of heterogeneity in the socioeconomic status of Chilean municipalities and the frequent changes in their lockdown status allows investigating how the socioeconomic characteristics of their inhabitants affect observable measures of social sentiment, although these measures are generally observable for the population as a whole and not for specific segments of it. For sentiment analysis we resort to Twitter queries to gauge the social sentiment toward government interventions and to Google Trends to assess the interest that users have in topics related to the pandemic. We perform our analysis using event study methods and panel data models similar to the difference-in-difference methodology.

Regarding Twitter, we find that abnormal sentiment responses are negatively related to increases in the number of people under lockdown, but with their statistical significance and economic effects concentrated among the wealthiest cohorts of the population, which suggests the existence of socioeconomic segregation among users of this platform. Furthermore, our results suggest that said Twitter socioeconomic segregation mirrors stock market segregation. Finally, regarding the intensity of Google searches for pandemic-related issues, a higher intensity is observed when a larger proportion of the total population is under lockdown, but with no discernible differential effect for the wealthier cohorts of the population.

We have added to the current literature by providing evidence of socioeconomic segregation among Twitter users. This is an important result not only for academics but also for policymakers. As sentiment analysis is becoming a pervasive tool to evaluate the impact of economic and social policies, it should be considered whether observable social sentiment indicators reflect the feelings towards such policies of the population as a whole or those of specific groups. Moreover, our empirical approach, which hinges on the socioeconomic heterogeneity of Chilean municipalities and the dynamic features of the pandemic strategy, allows directly identifying the socioeconomic status of Twitter users, a rather hard task to achieve [16, 28, 59]. Additionally, and as a secondary result of our analysis, we demonstrate a substantial degree of socioeconomic segregation in stock market reactions to government announcements. Even though this result was mainly used to validate the wealth ranking of Chilean municipalities used in the sentiment analysis, it is a novel result. We have no knowledge of other studies that explain reactions of the stock market as a whole to exogenous shocks (government announcements) based on the socioeconomic—or any other features—of the population affected by these shocks.

Our results must be interpreted in light of a number of limitations. First, the socioeconomic variables used to sort the population, the *MPI* and municipal income, are measured at a

municipal level. Even though municipalities are the smallest administrative unit in Chile, and there is in fact a high degree of income-based urban geographical segregation, there is intra-municipal heterogeneity in the *SES* of the population which we are currently not able to capture.

Second, our social sentiment proxies have some limitations. Google Trends counts aggregate "searches" but does not identify those who perform them, so it is not possible to know whether a spike in the relative proliferation of a search term is due to a few power users or many infrequent users. Also, younger individuals are relatively more likely to use Google Search than older individuals [60]. Lastly, some of these 19 terms may change, in either direction, without a direct relation with government announcements. Regarding Twitter, this study is restricted to #COVID2019chile and #CoronaVirusEnChile; the choice of these hashtags limits the generalization of our findings, but this is equally true for any other selection criteria.

In any case, the results obtained are consistent between the two proposed empirical methodologies. Our results hold when considering controls and are robust to alternative *SES* rankings and classifications.

## Supporting information

**S1 Table. Chilean municipalities.** This table presents population information for the 120 Chilean municipalities with more than 13,000 inhabitants. Municipalities are sorted from low to high, according to their *MPI*.
(PDF)

**S1 File. The database.**
(ZIP)

## Acknowledgments

Thanks to Sebastian Gonzalez and Vicente Dourthe for outstanding research assistance in the data-building phase of this project.

## Author Contributions

**Conceptualization:** Fernando Díaz, Pablo A. Henríquez.

**Data curation:** Fernando Díaz.

**Formal analysis:** Fernando Díaz, Pablo A. Henríquez.

**Investigation:** Fernando Díaz, Pablo A. Henríquez.

**Methodology:** Fernando Díaz, Pablo A. Henríquez.

**Visualization:** Fernando Díaz, Pablo A. Henríquez.

**Writing – original draft:** Fernando Díaz, Pablo A. Henríquez.

**Writing – review & editing:** Fernando Díaz, Pablo A. Henríquez.

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
