## [Decision Letter · Decision Letter 0]

1 Mar 2021

PONE-D-21-01436

Twitter and Google Trends Sentiment in a highly segregated Economy: The COVID-19 Case in Chile

PLOS ONE

Dear Dr. Henríquez,

Thank you for submitting your manuscript to PLOS ONE. After careful consideration, we feel that it has merit but does not fully meet PLOS ONE’s publication criteria as it currently stands. Therefore, we invite you to submit a revised version of the manuscript that addresses the points raised during the review process.

The paper approaches a very topical subject, but many drawbacks befall. For instance, the discussion regarding prior literature should be expanded, the quantitative framework should be improved, whereas the practical implications of the research should be noticed. Besides, the English language should be improved.

We look forward to receiving your revised manuscript.

Kind regards,

Stefan Cristian Gherghina, PhD. Habil.

Academic Editor

PLOS ONE

Journal Requirements:

2. In your Methods section, please include additional information about your dataset and ensure that you have included a statement specifying whether the collection method complied with the terms and conditions for the website.

3.In your Data Availability statement, you have not specified where the minimal data set underlying the results described in your manuscript can be found. PLOS defines a study's minimal data set as the underlying data used to reach the conclusions drawn in the manuscript and any additional data required to replicate the reported study findings in their entirety. All PLOS journals require that the minimal data set be made fully available. For more information about our data policy, please see http://journals.plos.org/plosone/s/data-availability.

Reviewers' comments:

Reviewer's Responses to Questions

**Comments to the Author**

1. Is the manuscript technically sound, and do the data support the conclusions?

Reviewer #1: No

Reviewer #2: Partly

Reviewer #3: Yes

Reviewer #4: Yes

Reviewer #5: Partly

Reviewer #6: Yes

2. Has the statistical analysis been performed appropriately and rigorously? 

Reviewer #1: No

Reviewer #2: No

Reviewer #3: Yes

Reviewer #4: I Don't Know

Reviewer #5: N/A

Reviewer #6: I Don't Know

3. Have the authors made all data underlying the findings in their manuscript fully available?

Reviewer #1: Yes

Reviewer #2: No

Reviewer #3: Yes

Reviewer #4: Yes

Reviewer #5: Yes

Reviewer #6: Yes

4. Is the manuscript presented in an intelligible fashion and written in standard English?

Reviewer #1: Yes

Reviewer #2: Yes

Reviewer #3: No

Reviewer #4: Yes

Reviewer #5: Yes

Reviewer #6: Yes

5. Review Comments to the Author

Reviewer #1: This paper analyzes the impact of changes in the quarantine status of Chilean municipalities on stock market returns, trends in Google searches, and sentiment as inferred by activity on Twitter. The main results are that events in which richer municipalities enter (exit) quarantine are associated with negative (positive) reactions on the stock market and on sentiment (as inferred via Twitter activity), and that increases in the total amount of population under quarantine are associated to increases in COVID-related search activity on Google.

I think the paper combines an important question with an interesting empirical setting and carefully collected data. However, I have some major concerns on the measurement, the empirical analysis, and the interpretation of the results.

1) The interpretation of the results is pushed in the direction of inferring degrees of “segregation” among users of Google search or Twitter. I find this interpretation not fully supported and, in any case, not central to the message of the paper. If the question of the paper is to determine the reaction of Google and Twitter users to changes in the quarantine status of different municipalities, why shall we try to address the unrelated question of users’ selection into those platforms?

2) Similarly, stock market segregation is mentioned in the abstract and the paper, but never properly defined. The statement that stock market segregation mirrors socio-economic segregation, and that this result can be derived from the finding on stock market returns should be clarified or better explained. In any case, I think this statement doesn't belong neither to the abstract nor to the introduction. Related to this point, I would de-emphasize the claim in the last paragraph of the Literature Review and the last paragraph before the conclusions.

3) The authors use the Multidimension Poverty Index (MPI) as a proxy for socio-economic status. Why not using more standard measures such as average per capita income? Related to this point, how did the authors select the 12% threshold to define municipalities in the “wealthy” group? I think it is important to show at least some robustness around these measures and definitions.

4) How are the (-1,0) and the (-1,+1) windows defined? Is 0 defined as the day of the announcement? It is critical to properly justify the transformation of the outcome variables in “cumulative” terms between t1 and t2. It is unclear why cumulating is the right transformation (instead, for example, of taking the difference between t2 and t1). In general, event-study settings should isolate the impact of the event by comparing a pre-event outcome with a post-event outcome, but my understanding of the empirical setting is that the dependent variables are averages of the pre- and post-event outcomes.

5) What are the “Sentiment Controls” used in Equation (13)? Are they the same as the “Sentiment Score” defined in Equation (8)? Why are they included in Equation (13)? Since sentiments are likely to be themselves affected by quarantine decisions, including them as a control might make inference worse. I have the same concern about Equations (14) and (15), where CAR_t, an outcome in Equation (13), here appears as a control.

6) Although the paper is certainly understandable, the quality of the writing is sometimes not as good as it could be. There are multiple typos (e.g. Twitter is occasionally spelled “Tweeter”), and some sentences that should be rephrased (e.g. “are statistically insignificant from 0” I believe should be “are statistically insignificant”).

Reviewer #2: This is an insightful study of stock market and pandemic related sentiment reactions to lockdown announcements. Moreover, it manages to highlight the role of socioeconomic characteristics as a confounding factor. Therefore, I would like to congratulate the authors on the very topical and relevant paper. I greatly enjoyed reading the paper and the results could be impactful and informative for the journal’s readership. However, some methodological choices and caveats require further consideration.

These are my comments that I would like to ask the authors to address:

[...]

Please see the enclosed comments.

Reviewer #3: Comments to the Author

Dear authors

The corrections comments are based on the manuscript you were followed.

Concern # 1: The justification behind this manuscript is weak and the problem is not valid.

Concern # 2: The strategies below would emphasize the novelty of your findings.

• Highlight the gaps in the Introduction section and mention how your study is going to address any/ some of the gaps

• In the presented Discussion, discuss the findings of the previous studies and specifically mention what new observation or insight was generated through your study results.

• In the Conclusion section, clearly, mention how your study advances the knowledge in the field

Concern # 3: Reorganize the abstract to conclude:

(a) The overall purpose of the study and the research problems you investigated.

(b) The basic design of the study.

(c) Major findings or trends found as a result of the study.

(d) A brief summary of your interpretations and conclusions.

Concern # 4: The introduction needs to clarify the (1) motivation, (2) challenges, (3) contribution, (4) objectives, and (5) significance/implication. All the information (should be) presented in sequence idea.

Concern # 5: In literature review section, the comparative analysis should be pointed to testify that this study is more advanced than others. The comparative need to be illustrated through table consist of desire information among this study and others studies.

Concern # 6: The authors mentioned the contribution in Literature review Section (Our study also contributes to the literature on the socioeconomic status inference of social media hidden user characteristics, one of the most active fields in information retrieval. In this area, the following references are worth mentioning.). I think these confusing readers.

Concern # 7: Remove the duplicated sentences among sections. Summarize the article as much as possible.

Concern # 8: write the conclusion and consider the following comments:

- Highlight your analysis and reflect only the important points for the whole paper.

- Mention the benefits

- Mention the implication in the last of this section.

Concern # 9: The paper has language and grammar issues. Carefully revised the paper by [a native English speaker]/[a professional language editing service] to improve the grammars.

Please, carefully review the manuscript to resolve these issues.

Reviewer #4: Please include more info about sentiment analysis in abstract

please indicate about the novelty of this work in comparison with other Sentiment analysis work

in your first two paragraphs, u did not include a single reference till the third paragraph, include more references in the first two paragraphs.

before including about the Covid-19 and economy, please first discuss about the Covid-19 from medical causalities and death and statistics, then u can start talking about economy.

include more recent studies related to Covid-19 and sentiment analysis like the following

Sentiment analysis of nationwide lockdown due to COVID 19 outbreak: Evidence from India

Sentiment analysis and its applications in fighting COVID-19 and infectious diseases: A systematic review

Sentiment Analysis of COVID-19 tweets by Deep Learning Classifiers—A study to show how popularity is affecting accuracy in social media

Twitter sentiment analysis on worldwide COVID-19 outbreaks

introduce a section for preprocessing and discuss your preprocessing and its issues

introduce two sections for theoretical and practical significance of your work and indicate how your work is different from others

Reviewer #5: The manuscript technically sound, and the data support the conclusions. Authors should explain some of the results. I have some doubts about the methodology and some robustness check should be done.

Reviewer #6: 1) Include some official sources regarding information mentioned in the second and third paragraphs of the introduction.

2) Include the link of INE and CASEN survey

3) Include an official source about the 25 government announcements

4) Chile was the only country to implement a dynamic quarantine? Otherwise, you could comment/compare to other countries

5) Draw an object process diagram about the analyzes realized in the methodology section. It will help visualize all the steps

6) Why you did not test classification algorithms for Twitter classification?

7) Highlit with bold the most important values in table 1 / provide some visual graphic. Since it presents many values it is difficult to analyze them

8) Include a discussion section comparing your results with related work. I miss some comparisons with other works.

9) Add challenges and limitations in the conclusion section

10) some references have doi and others don´t

6. PLOS authors have the option to publish the peer review history of their article (what does this mean?). If published, this will include your full peer review and any attached files.

Reviewer #1: No

Reviewer #2: No

Reviewer #3: No

Reviewer #4: No

Reviewer #5: No

Reviewer #6: No

---

## [Author Response · Author response to Decision Letter 0]

21 Apr 2021

Dear,

Stefan Cristian Gherghina, Academic Editor,

Thank you for your comment. The literature review has been expanded. To keep the manuscript at a reasonable length, in Section 2, Table 1, we present a brief review of recent works related to sentiment analysis.

Regarding the quantitative methods, we have performed sensitivity anal- yses and conducted robustness checks to confirm the validity of our results. Specifically, in section 4.2, Table 3, Panel B, we present results for the cumulative abnormal returns of the IGPA index observed upon government announcements, considering an alternative sorting of municipalities, based on Municipal Income. We obtain similar results, both in magnitude and statistical significance, to those obtained for the original MPI sorting. We have also added results obtained for different wealth cohorts of the popula- tion, and not only for the ABC1 segment of the population, as suggested by some Reviewers. A similar sensitivity analysis have been performed for our sentiment proxies in tables 4 and 5 in Section 4.3.

Also regarding our quantitative framework, and as an alternative ap- proach to analyze the impact of quarantine announcements on market sen- timent, we consider the suggestion of one of the Reviewers and perform a panel data regression similar to the Difference in Difference (DiD) method- ology, which allows us to take advantage of the panel structure of our data, increasing considerably the sample size for estimation. Results are presented in Table 6.

With respect to the practical implications of our work, these are now discussed in the Introduction and in the Conclusions.

Finally, as regards to language, the revised document has been proofread by a professional proofreading service.

---

## [Decision Letter · Decision Letter 1]

2 Jun 2021

PONE-D-21-01436R1

Social Sentiment Segregation: Evidence from Twitter and Google Trends in Chile during the COVID-19 Dynamic Quarantine Strategy

PLOS ONE

Dear Dr. Henríquez,

Thank you for submitting your manuscript to PLOS ONE. After careful consideration, we feel that it has merit but does not fully meet PLOS ONE’s publication criteria as it currently stands. Therefore, we invite you to submit a revised version of the manuscript that addresses the points raised during the review process.

The revised version of the manuscript improved in a positive manner, but further revisions regarding empirical outcomes' discussion and concluding remarks are required.

We look forward to receiving your revised manuscript.

Kind regards,

Stefan Cristian Gherghina, PhD. Habil.

Academic Editor

PLOS ONE

Journal Requirements:

Reviewers' comments:

Reviewer's Responses to Questions

**Comments to the Author**

1. If the authors have adequately addressed your comments raised in a previous round of review and you feel that this manuscript is now acceptable for publication, you may indicate that here to bypass the “Comments to the Author” section, enter your conflict of interest statement in the “Confidential to Editor” section, and submit your "Accept" recommendation.

Reviewer #1: (No Response)

Reviewer #2: All comments have been addressed

Reviewer #3: All comments have been addressed

Reviewer #4: All comments have been addressed

Reviewer #5: All comments have been addressed

2. Is the manuscript technically sound, and do the data support the conclusions?

Reviewer #1: Partly

Reviewer #2: Partly

Reviewer #3: Yes

Reviewer #4: Yes

Reviewer #5: Yes

3. Has the statistical analysis been performed appropriately and rigorously? 

Reviewer #1: Yes

Reviewer #2: Yes

Reviewer #3: Yes

Reviewer #4: Yes

Reviewer #5: N/A

4. Have the authors made all data underlying the findings in their manuscript fully available?

Reviewer #1: Yes

Reviewer #2: Yes

Reviewer #3: Yes

Reviewer #4: Yes

Reviewer #5: (No Response)

5. Is the manuscript presented in an intelligible fashion and written in standard English?

Reviewer #1: Yes

Reviewer #2: Yes

Reviewer #3: Yes

Reviewer #4: Yes

Reviewer #5: (No Response)

6. Review Comments to the Author

Reviewer #1: In my previous report, I had raised concerns regarding measurement, interpretation, and exposition. The authors revised the paper along several dimensions, taking into account many of my concerns. Overall, the paper is now better written and more convincing. However, I believe some points are left unaddressed.

1) I still feel unsure about the choice of 12% as a threshold to define wealthy municipalities. The authors conduct some robustness around this number, which is certainly reassuring. However, the way this threshold is chosen is still puzzling to me. My understanding is that 12% is the share of people in ABC1. Is this share “geographical” in any way (besides the obvious fact that richer people will tend to reside in richer municipalities)? In other words, how are the 12% of people belonging to ABC1 connected to the 12% of people living in the richer municipalities?

2) The authors interpret their findings in terms of degree of “segregation” in financial markets and internet platforms. I still do not see why the response to lockdown decisions is necessarily informative of the degree of segregation by socioeconomic status. There can be other reasonable explanations for why the financial market reacts more strongly to lockdowns in richer municipalities. For example, it is possible that lockdowns in richer municipalities have a more pronounced impact on the overall economy, or that publicly traded firms tend to be located in richer municipalities, and are more likely to be affected by lockdowns in those municipalities. One could build analogous arguments to explain the heterogeneity in the results for the other measures. Alternative explanations should at least be considered, mentioned, and discussed. If the authors believe that the heterogeneity in the estimated effects for CASA and CASVA can only be reconciled by postulating different degrees of socioeconomic segregation for Twitter and Google users, this should be made explicit. It is possible that I am simply missing what the authors mean by “socioeconomic segregation” in this setting. In this case, I encourage the authors to provide more guidance to the reader and make the definition of this concept as clear as possible.

3) I agree with the choice of including the “Stringency Index” as a control. However, I still believe that controlling for CAR in Tables 4 and 5 makes the inference worse. The stock market response is an outcome of the lockdown announcements, so its inclusion as a control makes it harder to evaluate the actual treatment effect of the announcement on CASA and CASVA. In Angrist and Pischke (2009)’s terminology, this is a “bad control”, in the sense that it is itself an outcome of the experiment.

4) While the quality of the writing is now significantly improved compared to the previous version, there are still inconsistencies in the notation and typos. Here are a few typos I have found:

- In Equation (3), the sub-index “i” is in the right-hand-side, but not in the left-hand-side.

- In Equation (5), the sub-index “i” refers to “cohort”, while in Equation (3) it refers to the stock market index.

- The notation in Equations (10) and (12) displays a similar inconsistency.

- I believe “columns (7) to (8)” at line 502 should be “columns (7) to (12)”.

- At line 644, “expected sings” should be “expected signs”.

References

Angrist, J. and Pischke, J.-S. (2009). “Mostly harmless econometrics: an empiricists guide”. Princeton: Princeton University Press.

Reviewer #2: I would like to congratulate the authors on the substantial revision of their manuscript. You have adequately addressed my earlier comments and suggestions.

I do have two minor comments left:

- While I laude the authors in implementing an additional estimation (the DiD 'like' approach) to accommodate a larger sample size, 325 is still rather minimal. Hence, I advise the authors to stress this caveat explicitly in the final version of the paper.

- Regarding my earlier comment 3.a, I understand that the municipal level is the most granular available. Moreover, I also understand that restrictions were applied to the municipality as a whole. Nevertheless, this does not fully address my concern that inferring general socioeconomic correlations based on the rank of an entire municipality is somewhat of a stretch, because after all - as acknowledged by the authors - the municipal level is compounding large heterogeneities. Therefore, I urge the authors to come up with a more satisfactory explanation for why their conclusions are immune to this. Otherwise, the conclusions should be phrased in less strong terms. Simply not having the data seems rather unsatisfactory.

Reviewer #3: Dear authors

The revised manuscript considered all correction comments within the current version

regards

Reviewer #4: The authors addressed all my comments and concerns, I believe the manuscript now is suitable for publication

Reviewer #5: The authors made substantial changes to the manuscript, in accordance with the Reviewers’ comments, the text is more readable. I have only a minor comment regarding ambiguity.

The search volume intensity includes the following 19 terms: corona, OMS (WHO), virus, COVID-19, SARS, MERS, epidemia (epidemic), pandemia (pandemic), síntoma (symptom), infectado (infected), propagación (spread), brote (outbreak), distanciamento social (social distancing), restricción (restriction), cuarentena (quarantine), suspender (suspend), viajar (travel), encierro (lockdown) and mascarilla (face mask).

The authors claimed that even if there is ambiguity in the search of pandemic related words, this goes against the effect they intent to quantify. The reason is that if the search intensity of key words that are subject to ambiguity do not change around government announcements (because they are in fact ambiguous), the statistic in Eq 9 in the revised version of the paper would be downward biased.

My concern in that some of these terms may change without any relation with government announcements. Moreover, the change may be on the wrong direction. This is the case for “suspender” that can be use against different backgrounds far away from the COVID-19 crisis.

7. PLOS authors have the option to publish the peer review history of their article (what does this mean?). If published, this will include your full peer review and any attached files.

Reviewer #1: No

Reviewer #2: No

Reviewer #3: No

Reviewer #4: No

Reviewer #5: No

---

## [Author Response · Author response to Decision Letter 1]

15 Jun 2021

Thank you, we have addressed all the minor changes requested.

---

## [Decision Letter · Decision Letter 2]

1 Jul 2021

Social Sentiment Segregation: Evidence from Twitter and Google Trends in Chile during the COVID-19 Dynamic Quarantine Strategy

PONE-D-21-01436R2

Dear Dr. Henríquez,

We’re pleased to inform you that your manuscript has been judged scientifically suitable for publication and will be formally accepted for publication once it meets all outstanding technical requirements. The author(s) should consider the comments formualted by the second reviewer. As sell, English language checking is more than required.

Kind regards,

Stefan Cristian Gherghina, PhD. Habil.

Academic Editor

PLOS ONE

Additional Editor Comments (optional):

Reviewers' comments:

Reviewer's Responses to Questions

**Comments to the Author**

1. If the authors have adequately addressed your comments raised in a previous round of review and you feel that this manuscript is now acceptable for publication, you may indicate that here to bypass the “Comments to the Author” section, enter your conflict of interest statement in the “Confidential to Editor” section, and submit your "Accept" recommendation.

Reviewer #1: All comments have been addressed

Reviewer #2: (No Response)

Reviewer #3: All comments have been addressed

Reviewer #4: All comments have been addressed

2. Is the manuscript technically sound, and do the data support the conclusions?

Reviewer #1: Yes

Reviewer #2: Partly

Reviewer #3: Yes

Reviewer #4: Yes

3. Has the statistical analysis been performed appropriately and rigorously? 

Reviewer #1: Yes

Reviewer #2: Yes

Reviewer #3: Yes

Reviewer #4: Yes

4. Have the authors made all data underlying the findings in their manuscript fully available?

Reviewer #1: Yes

Reviewer #2: Yes

Reviewer #3: Yes

Reviewer #4: Yes

5. Is the manuscript presented in an intelligible fashion and written in standard English?

Reviewer #1: Yes

Reviewer #2: Yes

Reviewer #3: Yes

Reviewer #4: Yes

6. Review Comments to the Author

Reviewer #1: The authors have properly addressed all the comments in my previous report. Thank you.

I believe there is a typo in the header of Table 2, last column (Averge -> Average).

Reviewer #2: I would like to thank the authors for the additional clarifications they provided to my remaining questions. The paper has indeed improved again. However, I would like to encourage the authors to further convince me and the reader since the remaining points are imperative to the results.

1) First, I laude that the authors took to hearth my concern regarding the limited sample size, even after the new DiD approach. Moreover, they do well of adding the sentence in line 585, stating that the sample size remains limited. Nevertheless, this caveat comes quite late in the text and moreover sheds little light on the actual problems that may follow as a result of the caveat. For example, earlier results in the table (cf. Table 4 and 5) build on 25 observations only. The authors would do well in emphasizing that some of the assumptions underlying the estimation approaches (e.g. consistency of the estimator) may not hold and highlighting possible consequences. The risk of reporting false-positives being one of them, especially when reporting confidence levels as high as 85%.

2) Second, I would like to come back again to the inference of general socioeconomic correlation based on the rank of the entire municipality. The evidence on the variability among the richer municipalities is interesting. It does indeed underpin that the richer municipalities are more homogeneous in terms of the MPI. And, while no evidence per se is provided, they may also be more homogeneous in non-poverty related dimensions. Nevertheless, I do not see how this fully absolves the need to control for other predominant characteristics of these specific municipalities. Take for example the extreme case that richer households tend to have substantially more outgoing lifestyles – not necessarily due to economic status, but for example for health reasons they go more to parks, for runs, etc. In such cases, the richer municipalities’ sentiment may indeed be affected more negatively due to stay-at-home orders. In addition, this correlation would show up in your results, yet is not due to wealth but a confounding factor. Whereas I am inclined to believe the authors story from a purely theoretical point of view, I do feel that more reassurance is needed from a methodological point of view.

Reviewer #3: Dear authors

The revised manuscript considered all correction comments within the current version regards. The current version can be published now.

Reviewer #4: (No Response)

7. PLOS authors have the option to publish the peer review history of their article (what does this mean?). If published, this will include your full peer review and any attached files.

Reviewer #1: No

Reviewer #2: No

Reviewer #3: No

Reviewer #4: No

---

## [Editor Report · Acceptance letter]

5 Jul 2021

PONE-D-21-01436R2 

Social Sentiment Segregation: Evidence from Twitter and Google Trends in Chile during the COVID-19 Dynamic Quarantine Strategy 

Dear Dr. Henríquez:

I'm pleased to inform you that your manuscript has been deemed suitable for publication in PLOS ONE. Congratulations! Your manuscript is now with our production department. 

Kind regards, 

on behalf of

Dr. Stefan Cristian Gherghina 

Academic Editor

PLOS ONE